# Metabolomics in Multiple Sclerosis: Advances, Challenges, and Clinical Perspectives—A Systematic Review

**DOI:** 10.3390/ijms26189207

**Published:** 2025-09-20

**Authors:** Jan Smusz, Patrycja Mojsak, Paulina Matys, Anna Mirończuk, Joanna Tarasiuk, Kamil Grubczak, Aleksandra Starosz, Jan Kochanowicz, Alina Kułakowska, Katarzyna Ruszczyńska, Katarzyna Kapica-Topczewska

**Affiliations:** 1Department of Neurology, University Clinical Hospital in Bialystok, Marii Skłodowskiej-Curie 24A, 15-276 Bialystok, Poland; jansmusz3@gmail.com (J.S.);; 2Metabolomics and Proteomics Laboratory, Clinical Research Centre, Medical University of Bialystok, 15-276 Bialystok, Poland; 3Department of Regenerative Medicine and Immune Regulation, Medical University of Bialystok, Waszyngtona 13, 15-269 Bialystok, Poland; 4Klinika Bocian Fertility Clinic, ul. Akademicka 26, 15-267 Bialystok, Poland

**Keywords:** multiple sclerosis, metabolomics, immunometabolism

## Abstract

Multiple sclerosis (MS) is a chronic, immune-mediated neurodegenerative disorder marked by inflammation, demyelination, and neuronal loss within the central nervous system. Despite advances in diagnostics, current tools remain insufficiently sensitive and specific. Metabolomics has emerged as a promising approach to explore MS pathophysiology and discover novel biomarkers. This PRISMA-guided systematic review included 29 original studies using validated metabolomic techniques in adult patients with MS. Biological samples analyzed included serum, cerebrospinal fluid, and feces. Consistent metabolic alterations were identified across several pathways. The kynurenine pathway demonstrated a shift toward neurotoxic metabolites, alongside reductions in microbial-derived indoles, indicating inflammation and gut dysbiosis. Energy metabolism was impaired, with changes in glycolysis, tricarboxylic acid (TCA) cycle, and mitochondrial function. Lipid metabolism showed widespread dysregulation involving phospholipids, sphingolipids, endocannabinoids, and polyunsaturated fatty acids, some modulated by treatments such as ocrelizumab and interferon-β. Nitrogen metabolism was also affected, including amino acids, peptides, and nucleotides. Non-classical and xenobiotic metabolites, such as myo-inositol, further reflected host–microbiome–environment interactions. Several studies demonstrated the potential of metabolomics-based machine learning to distinguish MS subtypes. These findings highlight the value of metabolomics for biomarker discovery and support its integration into personalized therapeutic strategies in MS.

## 1. Introduction

Multiple sclerosis (MS) is a progressive, demyelinating autoimmune disease of the central nervous system. It is characterized by multifocal inflammatory lesions caused by T-lymphocytic and macrophage infiltrations as well as oligodendrocyte death. These pathological processes ultimately lead to the destruction of the myelin sheath and damage to nerve cells. B lymphocytes also play a significant role in the development of MS, as evidenced by increased intrathecal immunoglobulin synthesis, reflected by the presence of oligoclonal bands (OCB) in the cerebrospinal fluid (CSF) [1,2]. Pathologically, acute MS lesions display blood–brain barrier disruption, perivascular immune cell accumulation, and widespread demyelination accompanied by early axonal transection. Over time, these lesions evolve into either inactive gliotic scars or slowly expanding plaques with rims of activated microglia, w3hich are now recognized as drivers of disease progression. Importantly, the pathological burden extends beyond focal white matter lesions: cortical demyelination, diffuse microglial activation, mitochondrial dysfunction, and widespread axonal loss all contribute to irreversible neurodegeneration [3]. Clinically, these processes give rise to distinct phenotypes. Relapsing–remitting MS (RRMS) is characterized by episodes of acute inflammation and new lesion formation followed by partial remyelination. Secondary progressive MS (SPMS) emerges after years of RRMS, marked by diffuse neurodegeneration and gradual disability accumulation with fewer relapses. Primary progressive MS (PPMS) presents from onset with insidious neurological decline, often with spinal cord-predominant involvement. Clinically isolated syndrome (CIS) represents a first demyelinating event that may convert to definite MS, providing a unique window into the earliest stages of disease pathology [3,4]. The clinical presentation of MS is heterogeneous and reflects the multifocal distribution of demyelinating lesions within the CNS. The most common initial symptoms include limb weakness, optic neuritis with painful visual loss, paresthesias, diplopia and bladder dysfunction. Other frequent manifestations encompass impaired visual acuity, internuclear ophthalmoplegia, sensory loss with impaired vibration sense, spasticity, neuropathic pain, tremor, and ataxia. Cognitive deficits and depression are also common, affecting up to half of patients over the disease course, and contributing substantially to long-term disability and reduced quality of life [5].

MS is a global health issue, affecting an estimated 2.9 million people worldwide as of 2023. The disease is most common in high-income countries, particularly in North America and Western Europe [6]. Regardless of geographic location, women are approximately twice as likely as men to be affected by MS, exhibiting both higher prevalence and incidence [7,8]. MS is one of the most common diseases among young and middle-aged individuals and is a leading non-traumatic cause of disability in young adults [9,10].

The exact cause of MS remains unknown, although it likely involves a complex interplay of genetic predisposition, environmental triggers, and immune system dysregulation. Despite progress in diagnostic methods, challenges in early and accurate diagnosis still exist. Current approaches depend on clinical presentation, magnetic resonance imaging (MRI) findings, and CSF analysis, especially the detection of OCB [11]. However, these methods have limitations and may result in misdiagnosis, particularly in early or atypical cases. This underscores the ongoing need for new biomarkers that can enable earlier and more accurate diagnosis of MS. Additionally, there is a crucial demand for biomarkers that can monitor disease progression and assess therapeutic response in individuals receiving disease-modifying treatment (DMT).

In this context, metabolomics has emerged as a promising approach to explore the complex biochemical alterations associated with MS. This review aims to synthesize current knowledge by characterizing alterations in the levels of specific metabolites involved in key metabolic pathways whose disturbances contribute to the pathogenesis of MS, focusing on research published within the last five years. Additionally, it highlights recent advances in metabolomics technologies and discusses their potential for biomarker discovery in MS.

### Added Value Compared with Prior Reviews

Recent syntheses have either provided broad overviews of metabolomic alterations in MS [12], spotlighted arachidonic-acid-derived lipid mediators [13], or focused on NMR-based CSF/serum profiles across neurological diseases [14]. More recently, a review covered both human and animal studies with a strong emphasis on lipidomics, but without systematic methodology or integration with clinical outcomes [15]. Our review advances the field by updating the evidence base through August 2025; delivering a phenotype-stratified, multi-matrix integration (serum, CSF, feces, brain tissue) that captures directional trajectories across RRMS → SPMS → PPMS as detailed later in Tables and Figures of the Results and Discussion sections; explicitly linking metabolites to disability and neuroaxonal/astroglial injury metrics (EDSS/MSSS, sNfL, sGFAP) and to neuroinflammation quantified by TSPO-PET; synthesizing therapy- and lifestyle-induced metabolic shifts (ocrelizumab, fingolimod, IFNβ, progressive resistance training) [16,17,18,19,20,21,22,23,24,25]; and outlining translational applications, including candidate markers and machine-learning classifiers for clinical staging [26,27]. This integrative, clinically anchored synthesis provides decision-oriented perspectives that extend beyond compilation and are intended to support biomarker validation and translational application.

## 2. Methods

This systematic review was conducted in accordance with the PRISMA 2020 guidelines [28] as illustrated in Figure 1. A comprehensive literature search was performed to identify original studies reporting metabolomic alterations in patients with MS.

### 2.1. Search Strategy

A systematic literature search was conducted using keyword-based queries. The search terms included “metabolome in multiple sclerosis” and “metabolomics of multiple sclerosis”. The search was performed in the following databases: PubMed (54 articles retrieved) and Google Scholar (23 articles retrieved).

### 2.2. Eligibility Criteria

Only studies published within the last five years were included to focus on the most recent advances in the field. Studies were eligible if they met the following criteria: original, peer-reviewed research articles conducted in adult human populations (≥18 years of age), involving patients with clinically defined MS, reporting metabolomic outcomes based on biological samples (e.g., serum, CSF, feces), using validated analytical techniques (e.g., NMR, LC-MS, GC-MS), and published in English. Animal studies, in vitro studies, and research focusing exclusively on pediatric populations were excluded.

### 2.3. Information Sources

A comprehensive literature search of electronic databases was conducted using the US National Library of Medicine, PubMed (https://pubmed.ncbi.nlm.nih.gov/, accessed between 1 April and 1 June 2025) and Google Scholar (https://scholar.google.com/, accessed between 1 April and 1 June 2025).

### 2.4. Quality and Risk of Bias Assessment

Risk of bias and methodological quality of the included studies were not formally assessed, consistent with the objective of this review, which was to perform a descriptive synthesis of recent metabolomic studies in multiple sclerosis. Significant heterogeneity in study design, sample type (serum, CSF, feces, tissue), and metabolomic techniques (LC-MS, NMR, GC-MS) limited the feasibility of applying a unified risk-of-bias tool such as ROBINS-I or QUADAS-2. Therefore, the synthesis relied on qualitative integration rather than quantitative meta-analysis. Several recurrent methodological limitations should nonetheless be acknowledged. Some studies were based on very small patient groups, often enrolling fewer than 20 individuals [29,30,31], which limits statistical power and generalizability. In addition, demographic reporting was frequently incomplete: in some cases, age or sex distribution was not specified or was only partially reported [30,31,32]. Finally, variability in biological samples (serum, CSF, urine, feces, brain tissue) and analytical platforms (LC-MS, GC-MS, NMR, shotgun lipidomics) hampered direct comparisons across studies. These issues introduce a risk of bias and highlight that the observed metabolic alterations, although promising, should be interpreted as preliminary until validated in larger, standardized, multicenter cohorts.

### 2.5. Technical Validation and Quality Control

As part of our systematic assessment, we evaluated whether the included metabolomics studies reported details of technical validation and quality control (QC) procedures. We specifically screened for information on instrument calibration, the use of pooled QC samples or replicate injections, the addition of isotopically labeled internal standards, and adherence to the Metabolomics Standards Initiative (MSI) reporting framework.

Reporting was highly heterogeneous across studies. Several reports provided detailed procedures, including multi-point calibration curves with isotopically labeled internal standards and systematic QC injections [22,23,33,34], consistent with best practice, corresponding to MSI Level 1 identification. In contrast, many studies employed commercial platforms (e.g., Metabolon Inc., Nightingale NMR) that rely on validated compound libraries and recovery standards but did not disclose details of QC design or internal standards [21,35,36]. Finally, some untargeted NMR studies reported only chemical shift referencing and metabolite annotation based on HMDB or the literature, without the spiking of authentic standards, corresponding to MSI Level 2 identification [20,37].

Overall, while validated analytical platforms (LC-MS, GC-MS, NMR) were consistently used, the level of technical validation reporting varied substantially. Only a minority of studies explicitly described calibration protocols, QC sample design, or isotopically labeled standards, which limits reproducibility and cross-study comparability. This highlights the urgent need for harmonized standard operating procedures (SOPs) and transparent reporting of technical validation in future metabolomics research on multiple sclerosis.

### 2.6. Study Selection

A total of 76 records were identified through database searching. After removing 4 duplicates, 72 unique articles were screened based on their title and abstract. Of these, 11 were excluded due to irrelevance. Full-text retrieval was attempted for 61 articles, but 14 were not accesible.

The remaining 47 full-text articles were assessed for eligibility. After applying inclusion and exclusion criteria, 18 studies were excluded for the following reasons: review articles (n = 8), non-human or in vitro studies (n = 4), not related to MS (n = 2), not reporting metabolomic data (n = 3), and pediatric-only populations (n = 1).

Finally, 29 studies met the predefined criteria and were included in the qualitative synthesis.

### 2.7. Data Synthesis

The extracted data were synthesized qualitatively. Studies were grouped by MS phenotype (RRMS, SPMS, PPMS, CIS) and biological sample type (serum, CSF, feces, brain tissue). Both nominally significant alterations (*p* < 0.05) and those surviving multiple testing correction (FDR < 0.05) were reported; however, only FDR-significant findings were interpreted as robust. Table 1 summarizes the study characteristics, whereas Table 2 presents significantly altered metabolites together with reported *p*-values and FDR-adjusted values. Findings are further described narratively in the Results section. In this review, statistical significance was defined as *p* < 0.05 unless otherwise specified in the original study (e.g., when authors reported more stringent thresholds or applied multiple testing corrections such as FDR), as summarized in Table 2. No meta-analysis or quantitative pooling was conducted due to methodological heterogeneity.

## 3. Results

Metabolomics is an emerging data-driven field ofresearch that focuses on the comprehensive analysis of low-molecular-weight metabolites (<1.5 kDa) within biological systems. These molecules are end-products of numerous interconnected pathways, and their analysis provides a dynamic snapshot of both physiological and pathological states [46].

Two primary analytical approaches are commonly distinguished. Untargeted metabolomics aims to detect as many metabolites as possible in a biological sample, generating large datasets that capture global metabolic patterns. This strategy is particularly valuable in exploratory research, where it can reveal broad alterations and guide the focus of subsequent targeted investigations. In contrast, targeted metabolomics concentrates on a predefined set of compounds. It offers greater sensitivity and specificity, enabling detailed interrogation of selected pathways and serving as a tool for hypothesis validation [47].

Among analytical platforms, gas chromatography–mass spectrometry (GC-MS) and liquid chromatography–mass spectrometry (LC-MS) are the most frequently employed. GC-MS provides excellent resolution and well-established spectral libraries but requires chemical derivatization, which may introduce variability and limit the analysis of thermolabile compounds. LC-MS offers broader versatility, detecting polar, thermolabile, and higher-molecular-weight metabolites such as lipids, amino acids, and organic acids. Its strengths are high sensitivity and wide coverage, although inter-laboratory reproducibility can be more challenging due to variability in chromatographic conditions and data processing pipelines. Finally, nuclear magnetic resonance (NMR) spectroscopy provides a complementary, non-destructive approach. Although less sensitive than MS-based methods, NMR requires minimal sample preparation, delivers highly reproducible quantitative data, and allows direct analysis of intact biofluids, making it robust for comparative studies [48,49].

Metabolomic techniques provide a powerful window into the discovery of novel biomarkers for early pathological changes, disease progression, and therapeutic efficacy in disorders such as MS. Moreover, they provide insights into disease-altered metabolic processes, potentially revealing new targets for therapeutic intervention.

Studies analyzed in this review employed a range of analytical platforms, most commonly LC-MS-20, GC-MS-5, and NMR spectroscopy-8. The biological matrices analyzed included blood plasma or serum, CSF and others (urine, feces and, in a few cases, brain tissue). Detailed information on the biological samples analyzed in each study is presented in Table 1. Both untargeted and targeted metabolomics approaches were used across these studies.

Across the included studies, most cohorts were defined using the McDonald criteria, although different revisions were applied. Specifically, McDonald 2005 [33], McDonald 2010 [17,34,35,38,40,44,45], and McDonald 2017 [16,18,22,23,24,25,27,32,37,39,41,42,43] were reported. One study relied on the Poser criteria for CIS conversion [19]. Several reports did not specify diagnostic criteria and were therefore classified as not reported (NR) [20,21,29,30,36].

Additional diagnostic confirmation varied. MRI was used in multiple studies to assess demyelinating lesions, atrophy, or disease activity [17,22,24,25,31,32,37,39,40,41,42,43]. CSF analysis for oligoclonal bands (OCB) was performed in some cohorts [19,24,32,41]. More advanced modalities included PET (TSPO-PET) [20] and optical coherence tomography (OCT) [38,43]. A number of studies further strengthened diagnostic reliability by integrating biochemical biomarkers, such as serum neurofilament light chain (sNfL) [13,39], glial fibrillary acidic protein (GFAP) [13] and cytokine or immune profiling [18,20,44].

Overall, most studies consistently applied McDonald criteria (2010/2017 revisions), exceptions included the use of older criteria [19] or absence of clear reporting [20,21,29,30,36]. Diagnostic reliability was often reinforced through MRI, CSF analysis, and biochemical markers, and in selected studies by advanced imaging such as OCT or PET.

In the following sections, we discuss the main metabolic pathways found to be dysregulated in MS, including tryptophan–kynurenine metabolism, energy metabolism (glycolysis and the TCA cycle), lipid metabolism, amino acid and nitrogen metabolism, and xenobiotic metabolism. Each pathway is considered in terms of its biological relevance, the nature of the alterations observed, and the potential implications for biomarker discovery and therapeutic targeting in MS.

A comprehensive overview of significantly altered metabolites, with reported *p*-values and FDR-adjusted values where available, is presented in Table 2. This tabular synthesis complements the narrative description of pathway-level alterations provided in the main text.

### 3.1. The Kynurenine Pathway

Several studies [16,26,33,34,38,40,42,43] have reported significant alterations in the levels of kynurenine pathway (KP) metabolites, highlighting its involvement in the pathogenesis and progression of MS.

The KP is the primary route of tryptophan (TRP) catabolism and generates several bioactive metabolites with immunomodulatory and neuroactive properties. Among these, kynurenine (KYN), 3-hydroxykynurenine (3HK), anthranilic acid (AA), kynurenic acid (KYNA), 3-hydroxyanthranilic acid (3HAA), and quinolinic acid (QUIN) have been implicated in neurodegeneration. The rate-limiting step of TRP conversion to KYN in extrahepatic tissues is catalyzed by indoleamine 2,3-dioxygenase (IDO), whose expression is upregulated by proinflammatory cytokines such as interferon-gamma (IFN-γ). Several KP metabolites are neuroactive; for instance, QUIN is considered both neurotoxic and gliotoxic due to its excitotoxic effects on glutamatergic neurotransmission and its cytotoxicity toward neurons, astrocytes, and oligodendrocytes [50,51,52].

Targeted metabolomics has revealed altered circulating KP metabolites in MS: KYNA and 3HK are often decreased (↓ 1.2-fold and ↓ 1.5-fold, respectively [38]), while AA (↑ 3.1-fold [38]) and 3HAA [42] are elevated. Ratios reflecting enzyme activity (e.g., KYN:TRP, 3HK:KYN, QUIN:KYNA) are frequently disrupted, indicating shifts toward proinflammatory and neurotoxic states. Imaging studies further associate KYN and TRP levels with brain atrophy and choroid plexus volume [38].

Additional studies have shown altered TRP levels across MS subtypes. Serum TRP was elevated in RRMS compared to healthy controls (↑ 3.3-fold) and showed higher levels in RRMS than in progressive forms (SPMS and PPMS), with a trend toward stepwise decline across disease stages, reflecting progressive KP dysregulation [16]. In a Chinese cohort, L-TRP was decreased in both RRMS and PPMS and correlated negatively with tumor necrosis factor alpha (TNF-α) and positively with interleukins (IL-7, IL-12), Macrophage Inflammatory Protein-1 alpha (MIP-1α) and Monocyte Chemoattractant Protein-1 (MCP-1), suggesting compensatory immunoregulatory signaling [33]. Another study reported a similar pattern of reduced KYNA and elevated 3HAA, although findings did not reach statistical significance after correction for multiple comparisons [42]. Reduced KYN was associated with higher disability measured by the Expanded Disability Status Scale (EDSS), and other downstream metabolites, such as xanthurenic acid, correlated with retinal thinning in the GCIPL layer, linking KP alterations to neurodegenerative changes [44].

KP disturbances also extend to gut microbiota-derived TRP catabolites. Indolepropionate and its precursor indolelactate, produced by commensal bacteria, were reduced in MS serum, even though the abundance of indolepropionate-producing bacteria was unchanged. In contrast, indolelactate-producing taxa were markedly reduced, suggesting a microbial bottleneck that may impair neuroprotection and sustain inflammation [36]. Urine samples from patients with RRMS exhibited decreased KYN and KYN:TRP ratio, independent of treatment status or disease duration. Sex-specific patterns were also observed, with females exhibiting lower urinary TRP, KYN, AA, and serotonin. Elevated urinary TRP and indole-3-propionic acid (IPA) correlated positively with EDSS scores, whereas KYN:TRP correlated inversely. During relapse, increased AA and IPA, alongside a decreased KYN:AA ratio, indicated a dynamic metabolic response to inflammation [34]. Elevated L-TRP levels were also confirmed in RRMS compared to healthy controls, reinforcing the consistency of KP dysregulation across cohorts [26].

Recent evidence indicates that DMTs can modulate KP in MS, influencing both the balance between neurotoxic and neuroprotective metabolites and overall immune regulation. For example, fingolimod, a sphingosine-1-phosphate (S1P) receptor modulator, was associated with longitudinal changes in KP metabolism. Specifically, a gradual increase in serum tryptophan levels was observed over the two-year course of treatment, suggesting a shift in upstream TRP availability [40]. Collectively, these findings highlight a potential role of DMTs not only in immunoregulation but also in restoring metabolic homeostasis through modulation of the KP.

These findings collectively emphasize the kynurenine pathway as a crucial connection between immune dysregulation, neurodegeneration, and potential therapeutic targets in MS. Further targeted metabolomic studies are needed to clarify the clinical revelance of these effects across various MS treatments.

### 3.2. Energy Metabolism

Multiple studies have highlighted extensive alterations in energy metabolism in MS, including disruptions in glycolysis, mitochondrial function, and the utilization of alternative energy substrates.

#### 3.2.1. Ketone Bodies

Ketone bodies such as β-hydroxybutyrate (BHB), acetoacetate (AcAc), and acetone were elevated in patients with progressive MS compared with healthy controls, but not in RRMS. These metabolites positively correlated with clinical disability measures, including EDSS, Multiple Sclerosis Severity Score (MSSS), and ambulation time, suggesting increased reliance on lipid-derived energy in advanced disease stages [17].

#### 3.2.2. Central Carbon Metabolism: Glycolysis, Pyruvate, and TCA Cycle

Consistent alterations in central carbon metabolism have also been reported. Patients with RRMS showed increased levels of succinate (approximately 1.6-fold), adenosine triphosphate (ATP) (approximately 2.0-fold), and formate (approximately2.5-fold), while lactate was elevated in PPMS, possibly indicating enhanced anaerobic glycolysis [16]. However, in another recent study, succinic acid levels were reduced in both RRMS and PPMS compared to healthy controls [26]. Notably, a machine-learning-based analysis in the same study identified succinic acid as a key discriminatory metabolite, with lowest levels in PPMS, intermediate levels in RRMS, and highest levels in healthy controls. These findings contrast with earlier reports of succinate accumulation and highlight possible differences related to sample type, disease duration, or analytical approaches. Altogether, this underscores the need for further targeted studies to clarify the role of succinate and TCA cycle dysfunction in MS pathogenesis.

Treatment with ocrelizumab was associated with decreased levels of lactate and serine, suggesting modulation of glycolysis and serine–glycine metabolism [18]. In CSF from patients with CIS who later converted to MS, early elevations in glucose and lactate, accompanied by decreased creatine, were observed. Notably, glucose showed greater predictive value than OCB for MS conversion [19].

#### 3.2.3. Metabolic Reprogramming vs. Transcriptional Regulation

Broader metabolomic profiling confirmed enrichment of glycolysis, the tricarboxylic acid (TCA) cycle, and pyruvate metabolism in patients with RRMS, indicating systemic reprogramming of energy metabolism. However, transcriptomic analysis of peripheral blood mononuclear cells (PBMCs) did not reveal upregulation of canonical glycolytic or TCA cycle enzymes, suggesting post-transcriptional or systemic regulation [35].

Additional energy-related metabolites were altered in the plasma of patients with MS. Elevated levels of isocitric acid, sorbitol, O-phosphoethanolamine, and 4-oxoglutaramate correlated positively with TNF-α and IL-17, indicating a proinflammatory metabolic profile [33]. Conversely, acetate levels were decreased in patients with high brain inflammation measured by positron emission tomography (PET), consistent with mitochondrial impairment during neuroinflammation. Elevated serum glucose correlated with inflammation severity [20], and increased CSF hexoses were reported in SPMS [29].

#### 3.2.4. Inflammation-Linked Energy Metabolites and Mitochondrial Cofactors

Growing evidence indicates that disturbances in energy metabolism are closely intertwined with the pathophysiology and progression of multiple sclerosis (MS), affecting both peripheral and central systems. Alterations in pathways related to fatty acid catabolism, ketone body utilization, and mitochondrial function have been documented across disease stages. For instance, progressive resistance training (PRT) was associated with increased circulating levels of 3-hydroxyisobutyrate, 3-aminoisobutyrate, and glycerol, suggesting enhanced short-chain fatty acid catabolism [21]. Additional disturbances included decreased carnitine, a cofactor essential for β-oxidation of fatty acids in patients with MS, indicating impaired mitochondrial fatty acid transport and utilization [36]. Comparative metabolomic profiling between RRMS and SPMS revealed higher serum glucose and lower BHB in SPMS, consistent with reduced ketone body utilization in progressive disease [27]. In demyelinated brain tissue, reduced levels of energy-related compounds such as guanosine, pyridoxamine phosphate (a vitamin B6 derivative), and glutamate-γ-methyl ester were found not only within lesions but also in periplaque white matter, reflecting widespread mitochondrial and energetic dysfunction [30].

Collectively, these findings indicate that disturbances in energy metabolism are deeply intertwined with neuroinflammation, disease progression, and therapeutic response in MS.

### 3.3. Lipid Metabolism

Dysregulation of lipid metabolism has emerged as a central feature of MS pathophysiology, influencing immune signaling, neuroinflammation, membrane integrity, mitochondrial function, and gut–brain communication. A growing number of metabolomic studies [16,18,21,22,23,24,26,30,31,32,33,39,41,42,44,45,53], both targeted and untargeted, have identified alterations across diverse lipid classes, including polyunsaturated fatty acid (PUFA)-derived lipid mediators, sphingolipids, glycerophospholipids, endocannabinoids (eCBs), steroid hormones, short-chain fatty acids (SCFAs), acylcarnitines, and lipoproteins.

#### 3.3.1. Lipid Mediators in RRMS and Progressive MS

A consistent finding in RRMS is the altered composition of lipid mediators derived from omega-3 (ω-3) and omega-6 (ω-6) PUFAs. Elevated levels of linoleic acid (LA)-derived 9-hydroxyoctadecadienoic acid (9-HODE), γ-linolenic acid (GLA), and α-linolenic acid (ALA) have been associated with preserved white matter (WM) microstructure. Conversely, elevated 9,10-epoxy-12Z-octadecenoic acid (9,10-EpOME), a proinflammatory metabolite, correlated with reduced fractional anisotropy (FA), indicating WM injury. Notably, 9-hydroxyoctadecatrienoic acid (9-HOTrE), an ALA-derived lipid produced via 5-lipoxygenase (5-LOX), was higher in stable lesions and lower in active progression, suggesting a potential neuroprotective role during remission [39].

In progressive MS, lipid profiles showed a proinflammatory shift dominated by arachidonic acid (ARA)-derived species. Plasma levels of 5-HETE, 8-HETE, and 15-HETE were elevated and correlated with higher EDSS, sNfL, and brain atrophy (reduced total and deep gray matter volume). Simultaneously, LA-derived 9-HODE and 13-HODE were reduced. Omega-3 derivatives such as 14,15-DiHETE and 19,20-DiHDPA negatively correlated with GFAP, pointing to potential astroglial modulation [13].

#### 3.3.2. Effect of DMTs and Exercise on Lipid Metabolism

Disease-modifying therapies (DMTs) are pharmacological agents designed to alter the natural course of MS by modulating or suppressing the immune system. Unlike symptomatic treatments, DMTs aim to reduce the frequency of relapses, delay disability progression, and limit central nervous system (CNS) inflammation and demyelination [53].

DMTs influence lipid metabolism through diverse mechanisms, including immunomodulation, mitochondrial regulation, and oxidative stress response. Ocrelizumab, as shown by Siavoshi et al., significantly reduced plasma concentrations of steroid conjugates (e.g., pregnenediol disulfate, androstenediol disulfate), bile acids (e.g., taurocholenate sulfate), and lysophospholipids (e.g., 1-linoleoyl-GPA 18:2), indicating systemic effects on lipid homeostasis [17]. IFNβ modulates lipoprotein metabolism, particularly by reducing circulating very-low-density (VLDL) and low-density lipoprotein (LDL) particles. In the ABIRISK cohort, Waddington et al. showed that lipid-based biomarkers such as cholesteryl esters in VLDL, triacylglycerol/phosphatidylglycerol ratios, and free cholesterol predicted the development of anti-drug antibodies (ADA). Notably, lipid-lowering responses to IFNβ were more pronounced in ADA-negative individuals [45].

Exercise, particularly progressive resistance training (PRT), has been shown to beneficially alter lipid profiles. It was observed that PRT reduces circulating free fatty acids (e.g., myristate, palmitate, arachidonate) and modifies phospholipid composition. In parallel, levels of the neurosteroid dehydroepiandrosterone sulfate (DHEAS) increased, correlating with improved fatigue scores and maximal oxygen uptake (VO_2_ max) [21].

#### 3.3.3. Endocannabinoid System and Neurosteroids

Alterations in endocannabinoid system were disease- and sex-specific [23]. While canonical eCBs such as anandamide (AEA) and 2-arachidonoylglycerol (2-AG) were unchanged in group comparisons, CSF 2-AG was elevated in males and both AEA and 2-AG were elevated in younger patients with RRMS. These elevations were not reflected in serum, and central–peripheral correlations were weak. CSF ARA, a downstream product of AEA and 2-AG, was also increased in younger patients. Steroid profiling showed elevated CSF cortisol and corticosterone in SPMS and PPMS, along with increased 2-oleoylglycerol (2-OG) and prostaglandin E_2_ (PGE_2_), reflecting central lipid signaling. In serum, stearoylethanolamide (SEA) levels were weakly correlated with disease duration, while cortisol correlated negatively.

#### 3.3.4. Phospholipids, Sphingolipids, and Membrane Structure

Alterations in membrane lipid components included reduced serum levels of choline, O-phosphocholine, and sn-glycero-3-phosphocholine, indicating disturbed phospholipid turnover [16]. In a Chinese cohort, S1P and phytosphingosine were significantly decreased and correlated negatively with TNF-α and IL-17, and positively with IL-7, MIP-1α, and RANTES. Free fatty acids (palmitic, oleic, arachidonic acid) were elevated and correlated with inflammatory markers. Decreased levels of 17α-estradiol, a neurosteroid, were also observed [33].

In a unique study of 73 monozygotic twin pairs discordant for MS [44], ether-linked phosphatidylcholines (PC O-) and phosphatidylethanolamines (PE O-) were significantly reduced in MS-affected twins, alongside a decrease in conventional PC species. Lipid abnormalities were even more pronounced in untreated patients with MS, indicating a disease-related origin rather than a treatment effect. A separate lipidomic study in plasma of patients with PPMS [31] further confirmed the importance of membrane lipids. Among identified lipids, sphingomyelin SM (d18:1/14:0) and monohexosylceramide (MonoHexCer) (d18:1/20:0) emerged as the strongest discriminators between patients with PPMS and controls. Notably, higher SM (d18:1/14:0) levels correlated positively with cerebellar white matter volume, while increased MonoHexCer (d18:1/20:0) levels predicted accelerated brain atrophy. Lysophosphatidic acid (LPA-18:2) was also reduced in patients with rapidly progressing PPMS and SPMS, regardless of treatment status, and negatively correlated with clinical disability. These findings collectively highlight membrane lipid remodeling as a hallmark of MS, with implications for neurodegeneration and disease progression.

#### 3.3.5. Short-Chain Fatty Acids and Gut–Brain Crosstalk

Serum concentrations of acetate and derived ratios (e.g., acetate/butyrate) were lower in MS, particularly in untreated individuals. Patients with MS also showed reduced correlations between SCFAs and cytokines, suggesting impaired gut–immune communication. Propionic acid (PA) concentrations were decreased in both serum and feces. PA supplementation (1000 mg/day) restored regulatory T cell (Treg) function, reduced Th1/Th17 responses, stabilized EDSS, and lowered relapse rates. CSF PA levels increased after supplementation alongside basal ganglia volume on MRI [24,42].

#### 3.3.6. Early and Postmortem Lipid Alterations

In newly diagnosed, untreated patients, CSF linoleic and stearic acids were reduced, while oleic and palmitic acids were elevated, suggesting early proinflammatory lipid shifts [32,41]. Comparative studies showed shared lipid patterns in RRMS and PPMS, including increased levels of methyl 11,14-eicosadienoate and heptanoic acid in RRMS, with PPMS showing differences in myristic, trans-oleic, and pentadecanoic acids [26].

Postmortem analyses revealed increased sphingomyelin and ceramide in lesion cores and periplaque white matter. Sphingosine was elevated at lesion edges, while lysophospholipids, monoacylglycerols, and PUFA-linked eCBs were depleted. These changes correlated inversely with astrocyte and immune markers and positively with oligodendrocyte precursor cells, indicating a possible role in remyelination [30].

### 3.4. Nitrogen Metabolism

Dysregulation of nitrogen metabolism constitutes a core metabolic hallmark of MS, reflecting the interplay among inflammation, neurodegeneration, and immune cell activation. This complex metabolic domain includes perturbations in amino acids, peptides, polyamines, nitrogenous waste products, and nucleotides—each of which has been associated with disease activity, disability, and treatment response.

#### 3.4.1. Amino Acid Disturbances and Disease Phenotype

Multiple studies [16,33,42,43] have consistently reported decreased serum levels of several proteinogenic amino acids, including lysine, glycine, threonine, tyrosine, cysteine, serine, and glutamate, particularly in patients with RRMS compared with healthy controls. In contrast, pantothenic acid, methionine, and glutathione were increased in patients with RRMS, while creatine and alanine levels were lower in progressive MS subtypes [16]. In an independent cohort, L-TRP, L-tyrosine, L-phenylalanine, L-leucine, and L-isoleucine were also reduced, whereas L-glutamic acid and β-alanyl-L-arginine were elevated [33]. Notably, TRP levels negatively correlated with TNF-α and positively with IL-7, IL-12, MIP-1α, and MCP-1, underscoring its immunomodulatory relevance [33,42]. These amino acid disturbances appear to be functionally linked with disease activity and neurodegeneration. Altered concentrations of aromatic amino acids (AAAs) and branched-chain amino acids (BCAAs) were observed associated with EDSS scores and neuroretinal thinning. A reduction in AAA-derived microbial metabolites (e.g., phenyllactate, indolelactate, imidazolyllactate) was associated with monocyte-specific downregulation of aryl hydrocarbon receptor (AhR)-mediated signaling, reflecting diminished immunoregulatory tone in MS. Parallel increases in gut-derived, oxidized compounds such as p-cresol glucuronide, p-cresol sulfate, and phenylacetylglutamine were linked to higher disability, indicating proinflammatory amino acid fermentation in the dysbiotic gut [43].

#### 3.4.2. Amino Acids as Relapse and Disability Markers

Amino acid metabolism also demonstrated dynamic regulation in relation to disease activity. In a large prospective cohort of 201 patients with RRMS, lysine and asparagine were elevated during acute relapses and declined over time, whereas leucine and isoleucine showed an opposite trend, increasing during clinical stability. These four amino acids outperformed sNfL in predicting recent relapse activity (AUC = 0.911 vs. 0.575) and correlated with gadolinium-enhancing lesions on MRI, confirming their utility as metabolic relapse biomarkers [37]. In CSF, most amino acids—including glutamate, histidine, serine, arginine, tyrosine, and choline—were found to be reduced in newly diagnosed, treatment-naïve patients with MS, with some (e.g., histidine) correlating negatively with EDSS scores at 1 and 2 years [32,36].

#### 3.4.3. Polyamine and Biogenic Amine Metabolism

Interestingly, spermidine, a polyamine involved in immune cell proliferation and mitochondrial function, was increased in both serum and CSF, particularly in progressive MS, and positively correlated with IL-17 and TNF-α, highlighting its potential role in sustaining inflammation [25,33,41].

Biogenic amines including asymmetric dimethylarginine (ADMA), symmetric dimethylarginine (SDMA), and taurine were consistently detected in CSF samples, with ADMA levels elevated in patients with SPMS. Given its inhibitory action on nitric oxide synthase, increased ADMA may contribute to endothelial dysfunction and reduced cerebral perfusion [29].

#### 3.4.4. Peptide Metabolism

In brain tissue, lesion cores showed an accumulation of dipeptides and elevated activity in lysine and serine metabolic pathways, consistent with enhanced proteolysis and intracellular stress [30]. In serum from patients with RRMS, increased levels of Ser-Leu and 3-hydroxyhippurate were observed, while lanthionine, a thioether amino acid derivative, was elevated in both patients with RRMS and PPMS, suggesting altered microbial sulfur metabolism [26,36].

#### 3.4.5. Nucleotide Metabolism

Nucleotide metabolism was also notably disrupted. In brain tissue, purine nucleotide levels, particularly guanosine, were markedly reduced in demyelinated and periplaque regions, indicating impaired salvage or biosynthesis capacity [30]. In serum, inosine and nicotinamide adenine dinucleotide (NAD^+^) were increased in patients with RRMS compared to SPMS and PPMS, reflecting early redox and purine turnover changes [16]. Several studies reported decreased concentrations of uridine, deoxyuridine, pseudouridine, nicotinuric acid, and niacinamide, pointing to generalized impairment of nucleotide pool maintenance and RNA turnover [33,42]. Uridine, in particular, emerged as a highly discriminative feature in multivariate analysis [42].

Therapy-related metabolic shifts further emphasized nucleotide involvement. Treatment with ocrelizumab led to decreased 2′-deoxyuridine and increased *N*-acetyl-β-alanine, implicating the pyrimidine pathway in treatment response [18]. In parallel, a separate study identified enhanced activity in xanthine-related metabolic pathways, including caffeine degradation, and reduced xanthurenic acid, with these purine catabolism changes correlating with EDSS scores and neuroretinal integrity [43].

Additional insights were derived from carbohydrate-linked nucleotide precursors. Elevated levels of ribose, D-glucuronic acid γ-lactone, erythrose, and L-threose were found in both patients with RRMS and PPMS compared to controls. These intermediates are involved in the pentose phosphate pathway and nucleotide interconversion, suggesting enhanced nucleotide demand during inflammatory or repair processes [26].

These findings highlight nitrogen metabolism as a crucial hub linking immune dysregulation, neurodegeneration, and treatment response in MS.

### 3.5. Other Metabolites

Beyond canonical metabolic pathways, MS is associated with alterations in a range of non-classical, microbiota-derived, and xenobiotic metabolites [16,18,19,25,32,33,36,42]. Several studies have reported reduced serum concentrations of persistent xenobiotics including perfluorooctanesulfonate (PFOS) and perfluorooctanoate (PFOA) following treatment with ocrelizumab, a monoclonal antibody targeting CD20-expressing B cells. In contrast, levels of propyl 4-hydroxybenzoate sulfate, a phase II detoxification product derived from benzoate metabolism, were increased, suggesting immunotherapy-induced modulation of hepatic conjugation and excretion pathways [18].

Among endogenous compounds, myo-inositol, a polyol involved in phosphoinositide signaling and osmoregulation, emerged as a consistently altered metabolite. It was significantly reduced in both serum (↓ 0.36-fold) [16] and CSF of patients with MS, particularly in progressive subtypes. Immunologically, its levels were negatively correlated with proinflammatory cytokines TNF-α and IL-17, while showing positive correlations with IL-12, MIP-1α, and MCP-1, suggesting a regulatory role in immune homeostasis [19,33,42].

Metabolites associated with vitamin and antioxidant metabolism also showed disease-specific alterations. Elevated levels of pantothenic acid (vitamin B5) and ascorbate (vitamin C) were observed in patients with RRMS, whereas progressive forms showed depletion of formate, 2-dehydropantothenate, and D-glucuronic acid [16,42]. Additional reductions were reported for skatole, 5-hydroxyindoleacetic acid, trans-cinnamic acid, and homovanillic acid, aromatic compounds linked to microbial and hepatic metabolism.

Gut microbiota-derived metabolites were also implicated. p-cresol sulfate, a uremic toxin and bacterial metabolite structurally resembling myelin basic protein-like epitopes, was elevated in patients with MS and correlated with both neurological deterioration and microbiome composition, though not with specific bacterial taxa. These findings point to host–microbiota metabolic crosstalk as a factor in disease pathogenesis [25,36]. In parallel, reduced fecal levels of vitamin B-related compounds (e.g., nicotinate, phenyllactate, protoporphyrin IX) and bioactive lipids (e.g., *N*-oleoyltaurine) were observed, particularly in patients transitioning to SPMS, potentially reflecting loss of beneficial microbial functions [25].

Additional non-classical metabolites with diagnostic or mechanistic relevance included increased levels of formamide, 2,2-dihydroxyacetic acid, and 3-methyloctan-2-one. In contrast, 3-indolepropionic acid, a microbial antioxidant with neuroprotective potential, was decreased [26]. Of note, myo-inositol outperformed OCB as a CSF biomarker for MS conversion, demonstrating high specificity and predictive accuracy [19].

Overall, these findings highlight that non-classical metabolites and host–microbiota interactions contribute significantly to MS pathophysiology and hold promise as novel biomarkers.

### 3.6. Metabolomics as a Diagnostic Tool in Multiple Sclerosis

The application of metabolomics in patients with MS extends beyond mechanistic insights and pathway analysis. It offers promising avenues for biomarker discovery that may support disease classification and subtyping, while its role in early diagnosis remains to be validated in clinical settings. Recent studies have demonstrated that metabolite-based signatures can accurately differentiate between patients with RRMS and SPMS, underscoring the translational value of metabolomic profiling in clinical neurology and its potential as a supportive biomarker strategy.

In one study, a supervised orthogonal partial least squares–discriminant analysis (OPLS-DA) model demonstrated robust discrimination between patients with RRMS and SPMS, even under moderate pre-analytical variability such as delayed plasma separation and single freeze–thaw cycle. Although performance declined significantly in samples stored for more than five years at –80 °C, likely due to metabolite degradation, the model still performed well under standard conditions. Its high discriminatory accuracy under typical clinical handling supports its real-world applicability. These findings highlight both the diagnostic potential of metabolomics in patients with MS and the critical role of sample quality in ensuring biomarker fidelity [27].

Complementary results were obtained using machine-learning-based approaches. In particular, support vector machine (SVM) classifiers achieved up to 77% accuracy in distinguishing patients with RRMS from healthy controls based on selected metabolite panels. However, these models failed to generalize to patients with PPMS, reinforcing the concept of subtype-specific metabolic phenotypes in MS. Top-ranked features in these classification models included 2-ethylhexanoic acid, ribose, erythrose, and 3-indolepropionic acid—metabolites not classically linked to canonical pathways but capable of capturing disease-relevant systemic alterations. Stratification performance remained high across various algorithmic platforms, including partial least squares–discriminant analysis (PLS-DA), random forest, and SVM, lending further support to their robustness and potential for clinical deployment [26].

Although these models achieved 70–80% accuracy in distinguishing MS subtypes, their performance remains below the threshold required for clinical application. They should therefore be viewed as proof-of-concept tools rather than ready-to-use diagnostic assays.

Together, these data indicate that metabolomics-based classifiers can serve as sensitive tools for staging MS, particularly in distinguishing relapsing from progressive. Overall, these findings support metabolomics as an emerging research approach for biomarker discovery, which in the future could serve as an adjunct tool to established diagnostic strategies.

### 3.7. Subgroup Analysis by MS Phenotype

In CIS, changes included increased glucose and lactate, together with an early shift in the kynurenine pathway, marked by reduced KYNA [19,42].

In RRMS, consistent alterations were reported in energy metabolism. Reported changes included increases in succinate and formate [16], alongside broader enrichment of glycolysis, pyruvate metabolism, and the TCA cycle [26,35]. Lipid metabolism was also frequently implicated, particularly sphingolipids and phosphatidylcholines [16,33,35]. Dysregulation of SCFA, notably reduced propionic acid, was observed in serum of patients with RRMS [24]. Additionally, protective lipids, such as 9-HODE and ALA, were decreased [39]. RRMS cohorts also showed further changes in the kynurenine pathway, with decreased concentrations of KYN and KYNA and increased concentrations of tryptophan [16,40,46,47]. Perturbations in amino acid metabolism were repeatedly described, with decreases in arginine, histidine, and glutamate in CSF [32,41]. In RRMS groups, microbial indoles, including myo-inositol and indole-3-propionic, acid were decreased [16,19,26].

In SPMS, findings included elevated ketone bodies including BHB and AcAc [17], increased arachidonic acid cascade including HETE [22], and higher levels of spermidine [25] and ADMA [29]. Additionally, decreased levels of nucleotides including guanosine were reported in patients with SPMS.

In PPMS, reported changes involved osmolytes and microbial-derived metabolites, with lower myo-inositol and higher p-cresol sulfate [25,33]. Alterations in steroid hormones and neurosteroids (cortisol, corticosterone and related metabolites) were consistently elevated in CSF of PPMS compared RRMS and controls [23,33]. PPMS cohorts also showed shifts in lipid metabolism including decreased levels of specific dihydroxyceramides [31].

Overall, the available studies suggest that different MS phenotypes are associated with partially distinct metabolic changes. These observations underline the heterogeneity of reported findings and highlight the need for further work in larger, stratified cohorts.

## 4. Discussion

This systematic review integrates the most recent metabolomic studies in MS, highlighting convergent findings across heterogeneous designs and analytical platforms. Despite differences in cohorts, sample types, and methodologies, several metabolic domains show consistent alterations, which together delineate a trajectory of biochemical reprogramming as the disease progresses from CIS to RRMS and progressive forms (SPMS, PPMS). These consensus findings help refine the pathogenic interpretation of metabolomics in MS and support its potential as a tool for disease staging.

Across multiple independent cohorts, the kynurenine pathway consistently showed a shift towards neurotoxic metabolites, with reductions in KYN, KYNA, and 3HK, paralleled by increases in QUIN and 3HAA [16,34,38,43]. This imbalance reflects proinflammatory activation of indoleamine-2,3-dioxygenase and aligns with neurodegeneration via excitotoxic and gliotoxic mechanisms. Microbiota-derived tryptophan metabolites, including indole-3-propionate and indolelactate, were also decreased [36,50], reinforcing the concept of impaired gut–brain kynurenine metabolism. These alterations correlate with clinical disability and retinal thinning, supporting their utility as early indicators of neurodegenerative processes.

Perturbations in energy metabolism were among the most reproducible findings. CIS converters showed early elevations of glucose and lactate in CSF, surpassing oligoclonal bands in predictive value for MS conversion [19]. In RRMS, succinate, ATP, and formate were frequently elevated, although some cohorts reported reduced succinate, highlighting methodological heterogeneity [16,23]. In progressive MS, reliance on alternative substrates was evident: β-hydroxybutyrate and acetoacetate were consistently increased and correlated with EDSS and MSSS [17]. Reductions in acetate and carnitine further point to mitochondrial dysfunction and impaired fatty acid transport [18,20]. Altogether, these data outline a metabolic transition from glycolysis-driven immune activation in early stages to ketone body utilization and mitochondrial exhaustion in progression.

Lipidomic studies also revealed highly reproducible alterations. In RRMS, protective PUFA-derived mediators such as 9-HODE, gamma-linolenic acid, and alpha-linolenic acid were linked to preserved white matter integrity, whereas proinflammatory lipids such as 9,10-EpOME correlated with tissue injury [39]. In SPMS and PPMS, arachidonic acid-derived species (5-, 8-, and 15-HETE) were elevated and associated with EDSS, neurofilament light chain, and brain atrophy, while protective HODE and docosapentaenoic acid derivatives declined [13]. Sphingolipid abnormalities, including reduced sphingosine-1-phosphate and ether-linked phosphatidylcholines and phosphatidylethanolamines, were repeatedly reported in serum, CSF, and twin studies [18,34]. In brain tissue, lesion cores were enriched in ceramides and sphingomyelins, whereas lysophospholipids and endocannabinoids were depleted [30]. SCFAs, particularly propionate and acetate, were consistently reduced in serum and feces [24,42], with supplementation shown to restore Treg function and stabilize EDSS [24]. These convergent findings position lipid metabolism as a critical determinant of MS progression, integrating immune, mitochondrial, and gut–brain signaling.

Amino acid and nitrogen metabolism showed comparable convergence. Most studies identified reductions in essential and aromatic amino acids (lysine, tyrosine, phenylalanine, leucine, isoleucine, arginine, histidine, serine, glutamate), particularly in RRMS and newly diagnosed MS [16,32,33,34]. These deficits were linked to both immune dysregulation and retinal thinning [43]. Dynamic oscillations in amino acid levels correlated with disease activity: lysine and asparagine rose during relapses, while leucine and isoleucine increased during stability, outperforming neurofilament light chain as relapse biomarkers [37]. Progressive forms were characterized by elevations in spermidine, a polyamine associated with mitochondrial stress and proinflammatory cytokines IL-17 and TNF-α [25,33,41]. Elevated ADMA in SPMS further implicates impaired nitric oxide signaling and endothelial dysfunction [29]. Together, these patterns highlight amino acid metabolism as a sensitive marker of disease state and activity.

Nucleotide metabolism was also notably disrupted. In RRMS, inosine and NAD^+^ were elevated compared with SPMS and PPMS, reflecting compensatory redox and nucleotide turnover [16]. By contrast, uridine, deoxyuridine, pseudouridine, and guanosine were decreased across progressive phenotypes and in demyelinated brain tissue [30,33]. These findings point to nucleotide exhaustion as a feature of chronic neurodegeneration and loss of repair capacity.

Several studies further revealed convergent signatures of microbiota and xenobiotic metabolism. Elevated microbial fermentation products, including p-cresol sulfate, p-cresol glucuronide, and phenylacetylglutamine, were associated with disability and neuroinflammation [25,36,43]. Conversely, protective metabolites such as indolepropionate, skatole, and propionate were decreased [24,36,42]. Myo-inositol, a central polyol involved in osmotic and glial signaling, consistently declined in both serum and CSF, particularly in progressive subtypes [16,19,32], and outperformed oligoclonal bands in predicting conversion from CIS [19]. Treatment also influenced the xenobiotic profile, as ocrelizumab reduced circulating perfluorinated compounds (PFOS, PFOA) and altered bile acid and steroid conjugates [18]. These results underscore the host–microbiota–environment axis as a key regulator of MS progression.

The metabolomic alterations described in our review often parallel proteomic findings, underscoring shared pathophysiological mechanisms. For example, shifts in kynurenine pathway metabolites that correlate with proinflammatory cytokines such as IFN-γ and IL-17 [33,34,43] align with proteomic studies demonstrating elevated cytokine signatures in newly diagnosed RRMS [54]. Metabolomic changes in lipid mediators converge with proteomic markers of tissue damage: higher levels of arachidonic acid-derived HETEs reflect both increased disability and elevated sNfL [13,18], a well-established biomarker of axonal degeneration validated in large longitudinal cohorts [55], while protective ω-3 derivatives show negative correlations with sGFAP [13], recently shown to capture astroglial injury in MS [56] and further associated with disease subtype and MRI markers of severity [57]. Proteomics and metabolomics capture different but complementary dimensions of MS biology: whereas proteomics highlights immune- and neurodegeneration-related proteins, metabolomics reflects systemic biochemical shifts in energy and lipid metabolism. Combining these approaches not only broadens the scope of biomarker discovery but also increases the likelihood of identifying clinically relevant markers that can improve disease monitoring and therapeutic decision-making [58]. Recent multi-omics studies reinforce this convergence: one integrated biochemical, proteomic, and metabolomic analyses of CSF to predict clinical conversion from CIS to MS [19], while another combined metabolomic profiling with scRNA-seq to link altered aromatic amino acid metabolism to immune cell transcriptional programs [43]. Although not part of the present systematic review, additional work has integrated blood metabolomic and transcriptomic data to stratify patients with SM by disease severity, highlighting the broader potential of cross-layer omics approaches to expand mechanistic insight and biomarker discovery [59].

The included studies varied substantially in sample size, ranging from small exploratory cohorts (n < 20) to large multicenter analyses (n > 300), as summarized in Table 1. This heterogeneity inevitably affects the robustness and generalizability of the reported findings. Although we could not perform a formal sensitivity analysis given the qualitative design of this review, we observed that key metabolite alterations, such as shifts in energy metabolism and lipid pathways, were consistently replicated in larger cohorts [21,36,43], whereas some associations identified in smaller NMR-based studies may reflect preliminary signals requiring further validation.

Another limitation is the heterogeneity in reporting statistical significance across studies, with some reporting *p*-values at different thresholds and others applying FDR corrections. This inconsistency reduces direct comparability but was retained in our tables to ensure accuracy of the original reports.

Reporting of comorbidities and exclusion criteria was highly heterogeneous across the included studies. Only a minority provided clear information on how potential confounding conditions were managed. Notably, some investigations applied highly restrictive criteria by including only patients on a single therapy and excluding all chronic diseases and concomitant medications [40], or by excluding neuroinflammatory disorders and dementia from the control group [23]. Similarly, other studies explicitly excluded participants with autoimmune or neurological diseases and those on psychopharmacotherapy [32,41]. By contrast, most studies did not specify exclusion criteria beyond basic demographics or treatment status [16,17,21,22], although some adjusted for covariates such as BMI or smoking [25,45], or considered pseudo-relapses due to infections [37]. Overall, metabolic, psychiatric, and autoimmune comorbidities were rarely reported or systematically excluded, complicating the interpretation of metabolomic changes as disease-specific.

Taken together, the reviewed studies suggest that MS phenotypes may be associated with partly distinct metabolic alterations. CIS and early RRMS have most often been linked to changes in glycolysis, the kynurenine pathway, and amino acid availability. In RRMS, additional findings include reductions in protective lipid mediators and microbial metabolites. In SPMS, reported changes include elevations of ketone bodies, arachidonic acid derivatives, and polyamines, together with signs of nucleotide depletion. In PPMS, disturbances have been more frequently described in osmolytes and microbial-derived metabolites, reductions in specific ceramides, and increases in steroid hormones and neurosteroids. While these patterns point to possible phenotype-specific metabolic signatures, the evidence remains heterogeneous and limited, particularly for progressive forms. To illustrate these observations, we provide a schematic overview (Figure 2). Together, these additions clarify how metabolomics can not only deepen mechanistic understanding but also support staging diagnosis, complementing established biomarkers such as neurofilament light chain and MRI measures.

Although this review was focused on human metabolomic studies, several of the altered pathways we identified overlap with therapeutic targets under active investigation. For instance, mitochondrial dysfunction and impaired bioenergetics have motivated repurposing strategies such as metformin, which reduced inflammation, improved disease outcomes, and attenuated demyelination in EAE models [60]. Activation of Nrf2 signaling, mirrored in metabolomic data, underlies the therapeutic effect of dimethyl fumarate, which induces Nrf2 pathway activation in PBMCs and promotes regulatory immune phenotypes in patients with SM [61]. Similarly, an increase in astrocytic PGC-1α in active MS lesions may reflect an endogenous oxidative stress-mitigating response [62]. While a systematic evaluation of preclinical and clinical drug data exceeds this review’s scope, our findings provide mechanistic grounding that could guide translational research.

In contrast to prior reviews [12,13,14], which were either pathway-centric, lipid-restricted, or method-limited, our work integrates findings from multiple analytical platforms and biofluids, restricted to adult human MS cohorts, and stratified by phenotype. By aggregating consensus metabolic signals across RRMS, SPMS, and PPMS, and embedding them within clinical and imaging contexts, this review delivers a systematic, translational perspective that was not achieved by earlier analyses [15].

### Limitations

While this review provides a comprehensive synthesis of recent metabolomic findings in multiple sclerosis, several limitations must be acknowledged. Substantial heterogeneity in study designs, sample types (serum, CSF, feces, brain tissue), and metabolomic techniques (LC-MS, GC-MS, NMR) precluded direct cross-study comparisons and prevented a meta-analysis. The review was not registered in PROSPERO, and a formal risk of bias or quality assessment was not conducted, as the included studies were predominantly descriptive and methodologically diverse. Potential confounding factors such as diet, medications, and comorbidities were not consistently reported across studies and may have influenced metabolite levels. Given these constraints, a quantitative meta-analysis was not feasible; accordingly, we deliberately adopted a qualitative synthesis framework to integrate and interpret the metabolomic findings across studies. A further limitation is the lack of harmonized pre-analytical protocols across studies. Only a minority of reports explicitly described SOPs or provided details of sample handling (e.g., fasting state, time to processing, storage conditions, freeze–thaw cycles) [19,23,27,29,32,37,41]. The majority of studies reported metabolite data without specifying these critical pre-analytical steps, complicating direct cross-study comparability.

Pre-anal;itilac variability presents further limitation. Many included studies reported nominal *p*-values without systematic correction for multiple comparisons, which increases the risk of false positive results and limits the robustness of proposed biomarkers [16,18,25,27,31,32,33,38,39,40,41].

Technical validation reporting was similarly heterogeneous. Only a minority provided detailed descriptions of calibration protocols, internal standards, or QC procedures [22,23,33,34], while most offered limited or no information. This variability reduces reproducibility and hampers direct cross-study comparisons, highlighting the need for harmonized reporting standards in future metabolomics research in multiple sclerosis.

Sample size varied widely, ranging from very small exploratory cohorts to large multicenter analyses. Small sample sizes increase the risk of bias and reduce the reproducibility of findings, limiting the strength of cross-study comparisons.

Most included studies did not systematically report or exclude comorbidities, with only a few providing clear criteria [23,32,40,41]. This lack of standardization limits comparability and generalizability of the findings.

Another limitation is the absence of a formal quality assessment framework. Tools such as ROBINS-I or QUADAS-2 are not readily applicable to metabolomics, while adherence to the Metabolomics Standards Initiative (MSI) reporting guidelines was partial and inconsistent across studies. Some studies provided extensive QC and metabolite identification criteria [18,22,23,26,35,43,45], while the majority reported metabolite data without systematic reference to QC measures or MSI reporting levels. As a result, we did not implement a structured scoring system, but emphasize that future systematic reviews should incorporate tailored quality appraisal approaches to ensure reproducibility and minimize bias.

Finally the reproducibility of machine-learning-based biomarker discovery. Only two studies applied such approaches [26,27]. Reporting was heterogeneous: while one study employed rigorous validation (external tenfold cross-validation with repetition, permutation testing, and reporting of accuracy, sensitivity, and specificity) [27], most other attempts provided less detail on hyperparameter settings and external replication [26]. Table 3 summarizes these methodological aspects. These examples illustrate both the feasibility and the current variability of machine-learning applications in MS metabolomics, underscoring the need for standardized workflows, including transparent hyperparameter specification, validation schemes, and replication in independent cohorts.

## 5. Conclusions

Metabolomics offers a comprehensive lens through which to understand the biochemical complexity of MS. The metabolic domains most consistently affected—kynurenine catabolism, mitochondrial energetics, lipid signaling, amino acid turnover and gut-derived metabolite processing—align closely with known pathological features of MS, including immune activation, demyelination, and neurodegeneration.

The reproducibility of findings across cohorts and analytical platforms remains an obstacle to clinical translation. Further challenges include the standardization of protocols for sample collection, storage, and data analysis, as well as controlling for confounding variables such as diet, medications, and comorbidities.

To fully harness the diagnostic and therapeutic potential of metabolomics in MS, future efforts should focus on

Longitudinal, phenotype-stratified metabolomic studies that capture dynamic metabolic trajectories across disease stages, including conversion from CIS to MS and progression from RRMS to SPMS/PPMS;Validation of candidate biomarkers (e.g., myo-inositol, lysine, acetate) in multicenter cohorts using harmonized protocols;Integration of multi-omics platforms, including transcriptomics, microbiomics, and neuroimaging, to build mechanistic models of MS pathophysiology;While metabolomics-based machine-learning classifiers demonstrate promising discriminatory performance (typically 70–80% accuracy), these approaches are not yet sufficiently robust for clinical decision-making. Their current role should be regarded as exploratory, requiring validation in larger, multicenter cohorts before they can be translated into practice;Exploration of therapeutic modulation, assessing how metabolic pathways respond to DMTs (e.g., fingolimod, ocrelizumab, IFNβ) or interventions like resistance training and probiotic supplementation;To improve reproducibility and comparability of metabolomic findings, future guidelines should mandate transparent reporting of pre-analytical procedures and support the development of SOPs and pre-analytical validation studies;Future studies should adopt harmonized standard operating procedures and transparent reporting of calibration, QC design, and internal standards, ideally aligned with the Metabolomics Standards Initiative, to improve reproducibility and facilitate clinical translation in multiple sclerosis;Future studies should harmonize inclusion and exclusion criteria and systematically document comorbidities, medications, and lifestyle factors to minimize confounding and enhance reproducibility;With continued advances in analytical technologies and systems-level approaches, metabolomics shows considerable promise for advancing personalized medicine in MS, particularly through biomarker discovery that may eventually support diagnosis, monitoring of disease progression, and development of targeted therapeutic strategies. However, clinical validation in large, standardized cohorts is still required before translation into practice.

## Figures and Tables

**Figure 1 ijms-26-09207-f001:**
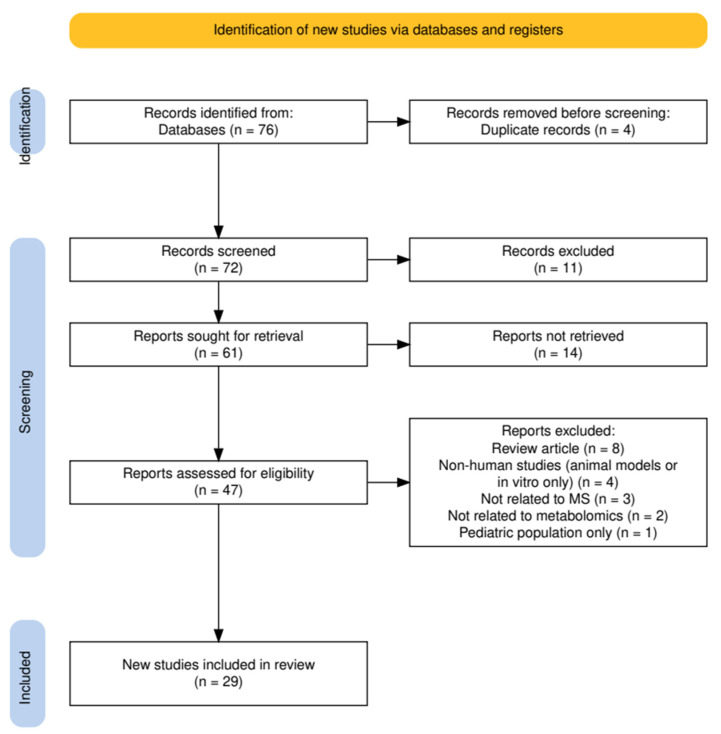
PRISMA flowchart.

**Figure 2 ijms-26-09207-f002:**
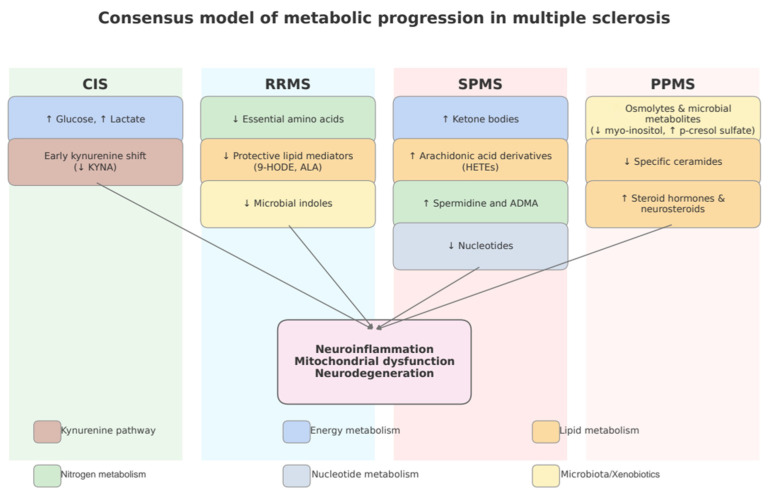
Consensus model of metabolic progression in multiple sclerosis. CIS most commonly shows increased glycolytic output (↑ glucose, ↑ lactate) and an early kynurenine pathway shift (↓ KYNA). RRMS additionally features ↓ essential amino acids, ↓ protective lipid mediators, and ↓ microbiota-derived indoles. SPMS is characterized by ↑ ketone bodies, ↑ arachidonic-acid derivatives (HETEs), ↑ spermidine and ADMA, and ↓ nucleotides. PPMS more often exhibits ↓ osmolytes together with microbial metabolites (↓ myo-inositol, ↑ p-cresol sulfate), ↓ specific ceramides, and ↑ steroid hormones/neurosteroids. Gray arrows indicate convergence toward shared processes—neuroinflammation, mitochondrial dysfunction, and neurodegeneration. Color coding: blue, energy metabolism; orange, lipid metabolism; green, nitrogen metabolism; beige, kynurenine pathway; slate, nucleotide metabolism; yellow, microbiota/xenobiotics. Patterns synthesize directionality reported across studies and do not imply causality; evidence for progressive forms remains comparatively limited.

**Table 1 ijms-26-09207-t001:** Main characteristics of the metabolomics studies included in this systematic review. Studies are grouped by first author and year, and summarized according to MS phenotype, cohort size, mean age (±SD or range where reported), sex distribution, biological sample type, analysis type (targeted or untargeted), and analytical technique.

Author (Year)	MS Phenotype	Number of Participants	Mean Age ± SD	Sex (F/M)	Biological Sample	Analysis Type	Analytical Technique
Staats Pires (2025) [38]	RRMS	RRMS—98 HC—39	RRMS: 42.3 (10.8) HC: 36 (6.3)	RRMS 72F/26M HC 28F/11M	serum	targeted	HPLC/GCLC
Wicks (2025) [17]	RRMS, PMS	RRMS—184 PMS—91 HC—152	RRMS: 44.2 (9.7) PMS: 54.1 (8.82) HC: 45.7 (13.9)	RRMS 139F/45M PMS 66F/25M HC 85F/67M	serum	targeted	NMR
Radford-Smith (2025) [20]	RRMS (progressors vs. non-progressors)	Non-progressors—52 Progressors—23	Non-progressors: 46.4 Progressors: 48.9	Non-progressors 43F/9M Progressors 16F/7M	serum	untargeted	NMR
Schwerdtfeger (2025) [25]	RRMS (progressors vs. non-progressors),	RRMS, stable—124 RRMS progression—68	RRMS, stable: 51.5 (9.8) RRMS progression: 54.1 (8.5)	RRMS stable 94F/30M RRMS progression 52F/16M	serum + feces	untargeted	HPLC
Alwahsh (2024) [16]	RRMS, PPMS, SPMS	RRMS—30 PPMS—30 SPMS—30 HC—30	RRMS: 37.0 (8.6) PPMS: 33.4 (7.0) SPMS: 37.9 (10.5) HC: 35.6 (2.5)	RRMS 19F/11M PPMS 15F/15M SPMS 15F/15M HC 15F/15M	serum	untargeted	NMR
Siavoshi (2024) [18]	RRMS (pre/post-ocrelizumab)	RRMS—31	40.84 (10.34)	21F/10M	serum	untargeted	LC-MS/CG-MS
Ladakis (2024) [30]	MS lesions	SPMS—5 Controls—6	Not reported	Not reported	brain tissue	untargeted	LC-MS + WGCNA
Fung (2024) [39]	RRMS	RRMS—28 HC—31	RRMS: 41.4 (9.1) HC: 41.9 (9.5)	RRMS 21F/7M HC 23F/8M	serum	targeted	HPLC-MS/MS
Datta (2024) [26]	RRMS, PPMS	RRMS—41 PMS—31 RRMS-HC—44 PPMS-HC—47	RRMS median: 39 (range 34–48) PPMS median: 49 (range 46–58) RRMS-HC median: 39.5 (range 33.45–39) PPMS-HC median: 53 (range 46.5–60.5)	RRMS 29F/12M PMS 22F/9M RRMS-HC 31F/13M PPMS-HC 35F/12M	serum	untargeted	2D GC × GC-MS
Meier (2024) [23]	RRMS, PPMS, SPMS	RRMS—58 SPMS—9 PPMS—4 Undefined—3 HC—80	MS: 43.6 (14.1) HC: 44.7 (16.1)	RRMS 42F/16M SPMS 6F/3M PPMS 2F/2M Undefined: 2F/1M HC 48F/32M	serum and CSF	targeted	LC-ESI-MS/MS
Murgia (2023) [40]	RRMS (fingolimod treatment)	RRMS—42 HC—22	MS: 39.0 (8.7) HC: 40.8 (13.8)	MS 23F/19M HC 17F/5M	plasma	untargeted	NMR
Broos (2023) [22]	RRMS, PMS	RRMS—170 PMS—115 HC—125	RRMS: 52.9 (0.9) PMS: 53.0 (0.9) HC: 52.9 (1.2)	RRMS 139F/31M PMS 65F/50M HC 92F/33M	serum	targeted	HPLC-MS/MS
Židó (2023) [32]	Early MS	MS—40 HC—33	MS median: 34 (range 18–54) HC: not reported	MS 31F/9M HC 26F/7M	CSF	untargeted/targeted	HPLC-MS/MS
Židó (2022) [41]	Early SM	MS—19 HC—19	MS: 36 HC: 35	MS 16F/3M HC 16F/3M	CSF	untargeted/targeted	HPLC-MS/MS
Zahoor (2022) [35]	RRMS	RRMS—35 HC—14	RRMS: 45 HC: 40	RRMS 22F/13M HC 9F/5M	serum	untargeted	UPLC-MS/GCLC/MS
Yang (2021) [33]	RRMS, PPMS	RRMS—20 PPMS—2 HC—21	MS: 34.8 (7.5) HC: 33.3 (8.5)	MS 14F/8M HC 13F/8M	plasma	untargeted	LC-MS/MS
Olsson (2021) [42]	RRMS/CIS	RRMS/CIS—58 HC—50	RRMS/CIS median: 34 (range 27–40) HC median: 33 (range 28–39)	RRMS/CIS 44F/14M HC 34F/16M	serum	targeted	UPLC-MS/MS,
Fitzgerald (2021) [43]	MS	MS—514 HC—241	MS: 42.54 (14.91) HC: 35.85 (15.71)	MS 468F/47M HC 224F/17M	plasma	untargeted	GC/MS, LC/MS/MS
Levi (2021) [36]	MS	MS—129 HC—58	MS: 38.3 (11.8) HC: 45.8 (12.5)	MS 93F/36M HC 29F/29M	serum	untargeted	LC/MS
Probert (2021) [19]	CIS converters vs. non-converters	Converters—22 Non-converters—32	Converters: 31.3 (9.9) Non-converters: 36.4 (11.2)	Converters 17F/5M Non-converters 21F/11M	CSF	untargeted	NMR,
Keller (2021) [21]	MS (PRT treatment)	MS—14 HC—13	MS: 42 (13) HC: 39 (14)	MS 12F/2M HC 11F/2M	serum	untargeted	UPLC-MS/MS
Yeo (2021) [37]	RRMS	RRMS relapse—28 RRMS last relapse in 1–6 months—28 RRMS last relapse in 6–24 months—34 RRMS last relapse in over 24 months—101	RRMS relapse: 38.3 (9.5) RRMS last relapse in 1–6 months: 38.7 (7.0) RRMS last relapse in 6–24 months: 43.5 (9.7) RRMS last relapse in over 24 months: 44.2 (9.9)	RRMS relapse 27F/11M RRMS last relapse in 1–6 months 23F/5M RRMS last relapse in 6–24 months 22F/12M RRMS last relapse in over 24 months 73F/28M	serum	untargeted/targeted	NRM
Yeo (2020) [27]	RRMS, SPSM	RRMS—31 SPMS—28	RRMS: 43.5 (9.7) SPMS: 58.1 (9.6)	RRMS 23F/8M SPMS 20F/8M	serum	untargeted	NMR
Gaetani (2020) [34]	RRMS	RRMS—47 HC—43	RRMS: 31.8 (9.7) HC: 32.7 (10.6)	RRMS 40F/7M HC 27F/16M	urine	targeted	HPLC-MS/MS
Carlsson (2020) [29]	SPMS	SPMS—12 HC—12	SPMS: 58.7 (7.5) HC: 54.2 (6.1)	SPMS 7F/5M HC: 7F/5M	CSF	targeted	LC-HRMS, FIA-HRMS
Duscha (2020) [24]	MS	RRMS—161 SPMS—103 PPMS—39 HC—68 NMO—1	RRMS: 45.8 (12.7) SPMS: 57.8 (9.4) PPMS: 60 (11.4) HC: 48.5 (14.5) NMO: 55	RRMS 102F/59M SPMS 62F/39M PPMS 16F/23M HC 32F/36M NMO 1M	serum, feces	targeted	LC-MS/MS
Penkert (2020) [44]	HC vs. MS/CIS (Monozygotic twins)	MS—73 HC—73	40.8 (12.1)	110F/36M	plasma	untargeted	shotgun lipidomics (LC-MS/MS)
Amatruda (2020) [31]	PPMS	PPMS—19 HC—8 Validation cohort: SPMS—11 RRMS—24	PPMS: 49.8 (11.1) HC: 42.1 (8.4) validation cohort: not reported	PPMS 11F/8M HC 4F/4M validation cohort-not reported	plasma	untargeted	LC-MS/MS,
Waddington (2020) [45]	RRMS (IFNβ-treated), CIS	ADA (−)—52 ADA (+)—30	ADA (−): 34.6 (9.3) ADA (+): 37.9 (9.8)	ADA (−) 37F/15M ADA (+) 19F/11M	serum, PBMCs	untargeted	NMR

**Table 2 ijms-26-09207-t002:** Summary of significantly altered metabolites in multiple sclerosis (MS). The table presents metabolomic alterations reported across studies included in this systematic review, grouped by MS phenotype (RRMS, SPMS, PPMS, CIS, early MS) and compared with healthy controls (HC) or between MS subgroups (e.g., progressors vs. non-progressors, ADA+ vs. ADA−, pre- vs. post-treatment). For each metabolite, the direction of change (↑/↓), comparison group, fold-change (where reported), *p*-value, and false discovery rate (FDR) are provided. Results denoted with (ns) were nominally significant (*p* < 0.05) but did not remain significant after multiple testing correction (FDR > 0.05). Entries with *p* = NR indicate that nominal *p*-values were not reported in the original study. Clinical correlations (e.g., EDSS associations) are also included where relevant.

Author (Year)	Clinical Groups	Metabolite	Direction	Fold-Change	Comparison	*p*-Value	FDR
Staats Pires (2025) [38]	RRMS, HC	KYNA	↓	1.2-fold (19%)	RRMS vs. HC	*p* < 0.05	FDR = 0.0394
		3HK	↓	1.5-fold (32%)	RRMS vs. HC	*p* < 0.05	FDR = 0.0008
		AA	↑	3.1-fold (212%)	RRMS vs. HC	*p* < 0.0001	FDR < 0.0001
		KYN/TRP	↑	1.15-fold (15%)	RRMS vs. HC	*p* < 0.01	FDR = 0.0187
		3HK/KYN	↓	1.5-fold (33%)	RRMS vs. HC	*p* < 0.001	FDR < 0.0001
		QUIN/KYNA	↑	1.27-fold (27%)	RRMS vs. HC	*p* < 0.05	NR
		3HAA/AA	↓	1.8-fold (44%)	RRMS vs. HC	*p* < 0.0001	FDR < 0.0001
Wicks (2025) [17]	RRMS, PMS and HC	β-hydroxybutyrate	↑	1.24-fold (24%)	RRMS vs. HC	NR	NR
		β-hydroxybutyrate	↓	0.87-fold (13%)	RRMS vs. PMS	NR	NR
		β-hydroxybutyrate	↑	1.43-fold (43%)	PMS vs. HC	*p* = 0.005	NR
		β-hydroxybutyrate	↑	1.15-fold (15%)	RRMS vs. RRMS	NR	NR
		Acetoacetate	↑	1.15-fold (15%)	RRMS vs. HC	NR	NR
		Acetoacetate	↓	0.88-fold (12%)	RRMS vs. PMS	*p* = NR	NR
		Acetoacetate	↑	1.31-fold (31%)	PMS vs. HC	*p* = 0.013	NR
		Acetoacetate	↑	1.13-fold (13%)	PMS vs. RRMS	*p* = NR	NR
Radford-Smith (2025) [20]	RRMS (progressors vs. non-progressors)	Glucose	↑	-	progressors vs. non-progressors	*p* = 0.0021 (discovery cohort); *p* = 0.0027 (validation cohort)	NR
		Glutamate	↑	-	progressors vs. non-progressors	*p* = 0.0006 (discovery cohort); *p* = 0.0409 (validation cohort)	NR
		Glutamine	↑	-	progressors vs. non-progressors	*p* = 0.0052	NR
Schwerdtfeger (2025) [25]	RRMS (progressors vs. non-progressors)	Spermidine (serum)	↑	-	progressors vs. non-progressors	*p* < 0.05	NR
		*p*-Cresol sulfate (serum)	↑	-	progressors vs. non-progressors	*p* < 0.05	NR
		Nicotinate (feces)	↓	-	progressors vs. non-progressors	*p* < 0.05	NR
		Phenyllactate (feces)	↓	-	progressors vs. non-progressors	*p* < 0.05	NR
		Protoporphyrin IX (feces)	↓	-	progressors vs. non-progressors	*p* < 0.05	NR
		*N*-Oleoyltaurine (feces)	↓	-	progressors vs. non-progressors	*p* < 0.05	NR
Alwahsh (2024) [16]	RRMS, PPMS, SPMS and HC	Tryptophan	↑	3.34-fold	RRMS vs. HC	*p* = 4.39 × 10^−7^	FDR < 0.05
		Succinate	↑	1.58-fold	RRMS vs. HC	*p* = 2.50 × 10^−6^	FDR < 0.05
		ATP	↑	1.98-fold	RRMS vs. HC	*p* = 1.11 × 10^−4^	FDR < 0.05
		Formate	↑	2.47-fold	RRMS vs. HC	*p* = 5.65 × 10^−4^	FDR < 0.05
		Inosine	↑	1.61-fold	RRMS vs. HC	*p* = 1.24 × 10^−3^	FDR < 0.05
		Histidine	↑	1.90-fold	RRMS vs. HC	*p* = 2.90 × 10^−3^	FDR < 0.05
		Glutathione	↑	1.47-fold	RRMS vs. HC	*p* = 3.29 × 10^−3^	FDR < 0.05
		Pantothenate	↑	1.37-fold	RRMS vs. HC	*p* = 5.54 × 10^−3^	FDR < 0.05
		Lysine	↓	0.48-fold	RRMS vs. HC	*p* = 3.11 × 10^−19^	FDR < 0.001
		Myo-inositol	↓	0.36-fold	RRMS vs. HC	*p* = 6.13 × 10^−18^	FDR < 0.001
		Glutamate	↓	0.57-fold	RRMS vs. HC	*p* = 1.56 × 10^−16^	FDR < 0.001
		Threonine	↓	0.38-fold	RRMS vs. HC	*p* = 7.64 × 10^−16^	FDR < 0.001
		Glycine	↓	0.45-fold	RRMS vs. HC	*p* = 9.51 × 10^−15^	FDR < 0.001
		Tyrosine	↓	0.61-fold	RRMS vs. HC	*p* = 3.33 × 10^−12^	FDR < 0.001
		Choline	↓	0.50-fold	RRMS vs. HC	*p* = 5.18 × 10^−12^	FDR < 0.001
		O-phosphocholine	↓	0.54-fold	RRMS vs. HC	*p* = 1.79 × 10^−9^	FDR < 0.001
		Serine	↓	0.71-fold	RRMS vs. HC	*p* = 7.47 × 10^−8^	FDR < 0.001
		Cysteine	↓	0.65-fold	RRMS vs. HC	*p* = 1.67 × 10^−3^	FDR < 0.001
		sn-Glycero-3-phosphocholine	↓	0.77-fold	RRMS vs. HC	*p* = 1.10 × 10^−6^	FDR < 0.05
		Creatine	↓	0.46-fold	RRMS vs. HC	*p* = 1.64 × 10^−5^	FDR < 0.001
		Tryptophan	↑	-	RRMS vs. PPMS	*p* = 1.42 × 10^−5^	FDR < 0.05
		Ascorbate	↑	-	RRMS vs. PPMS	*p* < 0.05	FDR < 0.01
		Histidine	↓	-	RRMS vs. PPMS	*p* = 5.48 × 10^−5^	FDR < 0.05
		Proline	↓	-	RRMS vs. PPMS	*p* = 8.03 × 10^−7^	FDR < 0.001
		Phenylalanine	↓	-	RRMS vs. PPMS	*p* = 1.24 × 10^−6^	FDR < 0.001
		Cysteine	↓	-	RRMS vs. PPMS	*p* = 3.93 × 10^−5^	FDR < 0.001
		Formate	↓	-	RRMS vs. PPMS	*p* = 1.49 × 10^−3^	FDR < 0.05
		Fumarate	↓	-	RRMS vs. PPMS	*p* = 2.33 × 10^−5^	FDR < 0.001
		Inosine	↑	-	RRMS vs. SPMS	*p* = 5.23 × 10^−3^	FDR < 0.05
		NAD^+^	↑	-	RRMS vs. SPMS	*p* = 9.88 × 10^−3^	FDR < 0.05
Siavoshi (2024) [18]	RRMS (pre/post—ocrelizumab)	Pregnenediol disulfate	↓	-	pre- vs. post-ocrelizumab	*p* = 1.78 × 10^−7^	FDR = 1.90 × 10^−4^
		Pregnenetriol disulfate	↓	-	pre- vs. post-ocrelizumab	*p* = 5.29 × 10^−5^	FDR = 1.13 × 10^−2^
		Androstene-diol disulfate	↓	-	pre- vs. post-ocrelizumab	*p* = 1.15 × 10^−4^	FDR = 1.76 × 10^−2^
		Taurocholenate sulfate	↓	-	pre- vs. post-ocrelizumab	*p* = 2.43 × 10^−4^	FDR = 2.16 × 10^−2^
		1-Linoleoyl-GPA (18:2)	↓	-	pre- vs. post-ocrelizumab	*p* = 2.39 × 10^−4^	FDR = 2.16 × 10^−2^
		2′-Deoxyuridine	↓	-	pre- vs. post-ocrelizumab	*p* = 1.64 × 10^−5^	FDR = 4.38 × 10^−3^
		PFOS	↓	-	pre- vs. post-ocrelizumab	*p* = 1.17 × 10^−6^	FDR = 6.23 × 10^−4^
		PFOA	↓	-	pre- vs. post-ocrelizumab	*p* = 2.20 × 10^−4^	FDR = 2.16 × 10^−2^
		Serine	↓	-	pre- vs. post-ocrelizumab	*p* = 1.43 × 10^−4^	FDR = 1.92
		Lactate	↓	-	pre- vs. post-ocrelizumab	*p* = 1.87 × 10^−4^	FDR = 2.16 × 10^−2^
		*N*-acetyl-β-alanine	↑	-	pre- vs. post-ocrelizumab	*p* = 6.85 × 10^−5^	FDR = 1.22 × 10^−2^
		Propyl	↑	-	pre- vs. post-ocrelizumab	*p* = 1.02 × 10^−5^	FDR = 3.64 × 10^−3^
		Propyl 4-hydroxybenzoate sulfate	↑	-	pre- vs. post-ocrelizumab	*p* = 1.02 × 10^−5^	FDR = 3.64 × 10^−3^
Ladakis (2024) [30]	MS lesions (SPMS) and HC	Sphingomyelins	↑	-	MS lesions vs. control	*p* = 1.26 × 10^−8^	FDR = 2.15 × 10^−7^
		Ceramides	↑	-	MS lesions vs. control	*p* = 1.26 × 10^−8^	FDR = 2.15 × 10^−7^
		Sphingosine	↑	-	MS lesions vs. control	*p* = 2.54 × 10^−6^	FDR = 2.88 × 10^−5^
		Dipeptides	↑	-	MS lesions vs. control	*p* = 0.04	FDR = 0.10
		Lysophospholipids	↓	-	MS lesions vs. control	*p* = 5.48 × 10^−9^	FDR = 1.86 × 10^−7^
		Monoacylglycerols	↓	-	MS lesions vs. control	*p* = 0.008	FDR = 0.04
		Hexosylceramides	↓	-	MS lesions vs. control	*p* = 0.002	FDR = 0.06
		Guanosine	↓	-	MS lesions vs. control	*p* = 0.00014	FDR = 0.001
		Pyridoxamine phosphate	↓	-	MS lesions vs. control	*p* = 0.00014	FDR = 0.001
		Glutamate-γ-methyl ester	↓	-	MS lesions vs. control	*p* = 0.00014	FDR = 0.001
		Nucleotides	↓	-	MS lesions vs. control	*p* = 0.01	FDR = 0.05
		UFAs	↓	-	MS lesions vs. control	*p* = 0.02	FDR = 0.06
		Endocannabinoids	↓	-	MS lesions vs. control	*p* < 0.001	FDR = 5.2 × 10^−9^
Fung (2024) [39]	RRMS and HC	9-HODE	↑	2.6-fold	preserved white matter integrity vs. white matter injury	*p* = 0.008	FDR ≈ 0.05
		ALA/GLA	↑	1.17-fold	preserved white matter integrity vs. white matter injury	*p* = 0.04	FDR ≈ 0.05
		9-HOTrE	↑	4.2-fold	preserved white matter integrity vs. white matter injury	*p* = 0.003	FDR ≈ 0.05
		9,10-EpOME	↑	-	white matter injury vs. preserved white matter	*p* = 0.01	FDR ≈ 0.05
		9-HOTrE	↓	-	white matter injury vs. preserved white matter	*p* = 0.003	FDR ≈ 0.05
Datta (2024) [26]	RRMS, PPMS and HS	2-Ethylhexanoic acid	↑	-	RRMS vs. HC	*p* < 0.05	FDR ≈ 0.3 (ns)
		Ribose	↑	-	RRMS vs. HC	*p* < 0.05	FDR ≈ 0.3 (ns)
		Erythrose	↑	-	RRMS vs. HC	*p* < 0.05	FDR ≈ 0.3 (ns)
		3-Indolepropionic acid	↓	-	RRMS vs. HC	*p* < 0.05	FDR ≈ 0.3 (ns)
		α-D-Glucopyranose	↑	-	RRMS vs. HC	*p* < 0.05	FDR ≈ 0.3 (ns)
		D-Glucuronic acid lactone	↑	-	RRMS vs. HC	*p* < 0.05	FDR ≈ 0.3 (ns)
		Heptanoic acid	↑	-	RRMS vs. HC	*p* < 0.05	FDR ≈ 0.3 (ns)
		L-Threose	↑	-	RRMS vs. HC	*p* < 0.05	FDR ≈ 0.3 (ns)
		Lanthionine	↑	-	RRMS vs. HC	*p* < 0.05	FDR ≈ 0.3 (ns)
		Linoleic acid	↑	-	RRMS vs. HC	*p* < 0.05	FDR ≈ 0.3 (ns)
		11,14-Eicosadienoic acid	↑	-	RRMS vs. HC	*p* < 0.05	FDR ≈ 0.3 (ns)
		Succinic acid	↓	-	RRMS vs. HC	*p* < 0.05	FDR ≈ 0.3 (ns)
Meier (2024) [23]	RRMS, PPMS, SPMS and HC	2-AG (CSF)	↑	-	Male RRMS vs. female RRMS and female HC	*p* < 0.05	FDR ≈ 0.05
		AEA (CSF)	↑	-	RRMS < 39 yrs vs. age/sex-matched HC	*p* < 0.05	FDR ≈ 0.05
		2-AG (CSF)	↑	-	RRMS < 39 yrs vs. age/sex-matched HC	*p* < 0.05	FDR ≈ 0.05
		AA (CSF)	↑	-	male vs. female RRMS	*p* < 0.05	FDR ≈ 0.05
		Cortisol (CSF)	↑	-	PMS (PPMS and SPMS) vs. RRMS and HC	*p* < 0.05	FDR ≈ 0.05
		Corticosterone (CSF)	↑	-	PMS (PPMS and SPMS) vs. HC	*p* < 0.05	FDR ≈ 0.05
		2-OG (CSF)	↑	-	RRMS ≥39 yrs vs. HC	*p* < 0.05	FDR ≈ 0.05
		PGE2 (serum)	↑	-	RRMS vs. HC	*p* < 0.05	FDR ≈ 0.05
		SEA (serum)	↑	-	MS (all subtypes) vs. HC	*p* < 0.05	FDR ≈ 0.05
Murgia (2023) [40]	RRMS (after fingolimod treatment) and HC	Alanine	↑	-	RRMS after fingolimod treatment vs. baseline	*p* < 0.05	FDR ≈ 0.05
		Phenylalanine	↑	-	RRMS after fingolimod treatment vs. baseline	*p* < 0.05	FDR ≈ 0.05
		Glycine	↑	-	RRMS after fingolimod treatment vs. baseline	*p* < 0.05	FDR ≈ 0.05
		Pyroglutamic acid	↑	-	RRMS after fingolimod treatment vs. baseline	*p* < 0.05	FDR ≈ 0.05
		Tryptophan	↑	-	RRMS after fingolimod treatment vs. baseline	*p* < 0.05	FDR ≈ 0.05
		Fructose	↑	-	RRMS after fingolimod treatment vs. baseline	*p* < 0.05	FDR ≈ 0.05
		Glucose	↑	-	RRMS after fingolimod treatment vs. baseline	*p* < 0.05	FDR ≈ 0.05
		2-Hydroxyisovalerate	↑	-	RRMS after fingolimod treatment vs. baseline	*p* < 0.05	FDR ≈ 0.05
		Creatinine	↑	-	RRMS after fingolimod treatment vs. baseline	*p* < 0.05	FDR ≈ 0.05
		Lactate	↓	-	RRMS after fingolimod treatment vs. baseline	*p* < 0.05	FDR ≈ 0.05
		Isoleucine	↓	-	RRMS after fingolimod treatment vs. baseline	*p* < 0.05	FDR ≈ 0.05
		Glutamate	↓	-	RRMS after fingolimod treatment vs. baseline	*p* < 0.05	FDR ≈ 0.05
		Lysine	↑	-	responders vs. non-responders	*p* < 0.05	FDR ≈ 0.05
		Lactate	↑	-	responders vs. non-responders	*p* < 0.05	FDR ≈ 0.05
		Glucose	↑	-	non-responders vs. responders	*p* < 0.05	FDR ≈ 0.05
Broos (2023) [22]	RRMS, PMS, and HC	11,12-DiHET	↑	-	RRMS vs. HC	*p* < 0.05	FDR < 0.05
		DPAn-3	↑	-	RRMS vs. HC	*p* < 0.05	FDR < 0.05
		15-HETE	↑	-	PMS vs. HC	*p* < 0.05	FDR (ns)
		8-HETE	↑	-	PMS vs. HC	*p* < 0.05	FDR (ns)
		5-HETE	↑	-	PMS vs. HC	*p* < 0.05	FDR (ns)
		11,12-DiHETE	↑	-	PMS vs. HC	*p* < 0.05	FDR < 0.05
		20-HETE	↑	-	PMS vs. HC	*p* < 0.05	FDR < 0.05
		ARA	↑	-	PMS vs. HC	*p* < 0.05	FDR < 0.05
		DGLA	↑	-	PMS vs. HC	*p* < 0.05	FDR < 0.05
		AdA	↑	-	PMS vs. HC	*p* < 0.05	FDR < 0.05
		9-HODE	↓	-	PMS vs. HC	*p* < 0.05	FDR < 0.05
		13-HODE	↓	-	PMS vs. HC	*p* < 0.05	FDR < 0.05
		14,15-DiHETE	↓	-	PMS vs. HC	*p* < 0.05	FDR < 0.05
		19,20-DiHDPA	↓	-	PMS vs. HC	*p* < 0.05	FDR < 0.05
Židó (2023) [32]	Early MS and HC	Arginine	↓	-	Early MS vs. HC	*p* = 0.0037	FDR < 0.05
		Histidine	↓	-	Early MS vs. HC	*p* = 0.0058	FDR < 0.05
		Glutamate	↓	-	Early MS vs. HC	*p* = 0.0145	FDR < 0.05
		Choline	↓	-	Early MS vs. HC	*p* = 0.0233	FDR < 0.05
		Tyrosine	↓	-	Early MS vs. HC	*p* = 0.0313	FDR < 0.05
		Serine	↓	-	Early MS vs. HC	*p* = 0.0473	FDR < 0.05
		Methionine	↑	-	Early MS vs. HC	*p* < 0.001	FDR < 0.05
		Linoleic acid	↓	-	Early MS vs. HC	*p* = 0.001	FDR < 0.05
		Stearic acid	↓	-	Early MS vs. HC	*p* = 0.029	FDR < 0.05
		Spermidine	↑	-	Early MS vs. HC	*p* = 0.0124	FDR < 0.05
		Oleic acid	↑	-	Early MS vs. HC	*p* = 0.015	FDR < 0.05
Židó (2022) [41]	Early MS and HC	Arginine	↓	-	Early MS vs. HC	*p* = 0.007	FDR < 0.05
		Histidine	↓	-	Early MS vs. HC	*p* = 0.012	FDR < 0.05
		Palmitic acid	↑	-	Early MS vs. HC	*p* = 0.039	FDR ≈ 0.05
Zahoor (2022) [35]	RRMS and HC	Phosphoethanolamine	↑	-	RRMS vs. HC	*p* < 0.05	FDR < 0.10 (ns)
		Lactic acid	↑	-	RRMS vs. HC	*p* < 0.05	FDR < 0.10 (ns)
		Fumaric acid	↑	-	RRMS vs. HC	*p* < 0.05	FDR < 0.10 (ns)
		3-Hydroxybutyrate	↑	-	RRMS vs. HC	*p* < 0.05	FDR < 0.10 (ns)
		Oleoylethanolamide (OEA)	↑	-	RRMS vs. HC	*p* < 0.05	FDR < 0.10 (ns)
		Sphingosine-1-phosphate (S1P)	↑	-	RRMS vs. HC	*p* < 0.05	FDR < 0.10 (ns)
Yang (2021) [33]	RRMS, PPMS and HC	L-Tyrosine	↓	-	RRMS and PPMS vs. HC	*p* < 0.001	FDR < 0.05
		L-Tryptophan	↓	-	RRMS and PPMS vs. HC	*p* = 0.015	FDR < 0.05
		L-Phenylalanine	↓	-	RRMS and PPMS vs. HC	*p* = 0.033	FDR < 0.05
		L-Leucine	↓	-	RRMS and PPMS vs. HC	*p* = 0.0049	FDR < 0.05
		L-Isoleucine	↓	-	RRMS and PPMS vs. HC	*p* < 0.001	FDR < 0.05
		Sphingosine-1-phosphate	↓	-	RRMS and PPMS vs. HC	*p* < 0.001	FDR < 0.05
		Sphinganine-1-phosphate	↓	-	RRMS and PPMS vs. HC	*p* < 0.001	FDR < 0.05
		Phytosphingosine	↓	-	RRMS and PPMS vs. HC	*p* < 0.01	FDR < 0.05
		17α-Estradiol	↓	-	RRMS and PPMS vs. HC	*p* < 0.001	FDR < 0.05
		Methyl jasmonate	↑	-	RRMS and PPMS vs. HC	*p* < 0.001	FDR < 0.05
		Myo-inositol	↓	-	RRMS and PPMS vs. HC	*p* < 0.001	FDR < 0.05
		Oleic acid	↑	-	RRMS and PPMS vs. HC	*p* = 0.046	FDR < 0.05
		Palmitic acid	↑	-	RRMS and PPMS vs. HC	*p* < 0.01	FDR < 0.05
		Arachidonic acid	↑	-	RRMS and PPMS vs. HC	*p* = 0.015	FDR < 0.05
		β-Alanyl-L-arginine	↑	-	RRMS and PPMS vs. HC	*p* < 0.01	FDR < 0.05
		4-Oxoglutaramate	↑	-	RRMS and PPMS vs. HC	*p* < 0.01	FDR < 0.05
		Isocitric acid	↑	-	RRMS and PPMS vs. HC	*p* < 0.05	FDR < 0.05
		O-Phosphoethanolamine	↑	-	RRMS and PPMS vs. HC	*p* < 0.01	FDR < 0.05
		Sorbitol	↑	-	RRMS and PPMS vs. HC	*p* < 0.01	FDR < 0.05
		Spermidine	↑	-	RRMS and PPMS vs. HC	*p* < 0.05	FDR < 0.05
		Homovanillic acid	↓	-	RRMS and PPMS vs. HC	*p* < 0.05	FDR < 0.05
		Deoxyuridine	↓	-	RRMS and PPMS vs. HC	*p* < 0.01	FDR < 0.0
		L-Arogenate	↓	-	RRMS and PPMS vs. HC	*p* < 0.05	FDR < 0.05
		Pseudouridine	↓	-	RRMS and PPMS vs. HC	*p* < 0.01	FDR < 0.05
		Uridine	↓	-	RRMS and PPMS vs. HC	*p* < 0.001	FDR < 0.05
		Dodecanoic acid	↓	-	RRMS and PPMS vs. HC	*p* < 0.05	FDR < 0.05
		Niacinamide	↓	-	RRMS and PPMS vs. HC	*p* < 0.05	FDR < 0.05
		α-Dimorphecolic acid	↓	-	RRMS and PPMS vs. HC	*p* < 0.01	FDR < 0.05
		cis-4-Hydroxy-D-proline	↓	-	RRMS and PPMS vs. HC	*p* = 0.004	FDR < 0.05
		*N*-Acetyl-L-asparagine	↓	-	RRMS and PPMS vs. HC	*p* < 0.01	FDR < 0.05
		L-Valine	↑	-	RRMS and PPMS vs. HC	*p* < 0.05	FDR < 0.05
		Glutamic acid	↑	-	RRMS and PPMS vs. HC	*p* < 0.001	FDR < 0.05
		Myristic acid	↓	-	RRMS and PPMS vs. HC	*p* < 0.05	FDR < 0.05
		DHEA	↓	-	RRMS and PPMS vs. HC	*p* < 0.05	FDR < 0.05
		Creatinine	↓	-	RRMS and PPMS vs. HC	*p* < 0.01	FDR < 0.05
		Nicotinuric acid	↓	-	RRMS and PPMS vs. HC	*p* < 0.05	FDR < 0.05
		Trans-cinnamic acid	↓	-	RRMS and PPMS vs. HC	*p* < 0.01	FDR < 0.05
		2-Dehydropantoate	↓	-	RRMS and PPMS vs. HC	*p* < 0.05	FDR < 0.05
Olsson (2021) [42]	RRMS, CIS and HC	Acetate	↓	-	RRMS and CIS vs. HC	*p* = 0.021	FDR = 0.067 (ns)
		Valine	↓	-	RRMS and CIS vs. HC	*p* = 0.005	FDR > 0.05 (ns)
		L-Methionine	↓	-	RRMS and CIS vs. HC	*p* = 0.025	FDR > 0.05 (ns)
		Kynurenic acid (KYNA)	↓	-	RRMS and CIS vs. HC	*p* = 0.019	FDR > 0.05 (ns)
		Creatine	↑	-	RRMS and CIS vs. HC	*p* = 0.022	FDR > 0.05 (ns)
		Pantothenic acid	↑	-	RRMS and CIS vs. HC	*p* = 0.021	FDR > 0.05 (ns)
		D-Glucuronic acid	↑	-	RRMS and CIS vs. HC	*p* = 0.031	FDR > 0.05 (ns)
		3-Hydroxyanthranilic acid (3HAA)	↑	-	RRMS and CIS vs. HC	*p* = 0.040	FDR > 0.05 (ns)
		Acetate/Butyrate ratio	↓	-	RRMS and CIS vs. HC	*p* = 0.005	FDR = 0.06 (ns)
		Acetate/(Propionate + Butyrate) ratio	↓	-	RRMS and CIS vs. HC	*p* = 0.010	FDR = 0.06 (ns)
Fitzgerald (2021) [43]	MS and HC	Phenyllactate (PLA)	↓	-	MS vs. HC	*p* < 1.0 × 10^−19^	FDR < 0.001
		3-(4-Hydroxyphenyl)lactate	↓	-	MS vs. HC	*p* < 1.0 × 10^−17^	FDR < 0.001
		Indolelactate	↓	-	MS vs. HC	*p* < 1.0 × 10^−15^	FDR < 0.001
		Imidazole lactate	↓	-	MS vs. HC	*p* < 1.0 × 10^−10^	FDR < 0.001
		Tyrosine	↓	-	MS vs. HC	*p* < 1.0 × 10^−6^	FDR < 0.001
		Tryptophan	↓	-	MS vs. HC	*p* < 1.0 × 10^−10^	FDR < 0.001
		Phenylpyruvate	↓	-	MS vs. HC	*p* < 1.0 × 10^−6^	FDR < 0.001
		Kynurenine	↓	-	MS vs. HC	*p* < 0.05	FDR ≈ 0.05
		Phenylacetylglutamine	↑	-	MS vs. HC	*p* < 1.0 × 10^−4^	FDR < 0.05
		*p*-Cresol glucuronide	↑	-	MS vs. HC	*p* < 1.0 × 10^−4^	FDR < 0.05
		*p*-Cresol sulfate	↑	-	MS vs. HC	*p* < 0.001	FDR < 0.05
		4-Hydroxyphenylpyruvate	↓	-	MS vs. HC	*p* < 1.0 × 10^−9^	FDR < 0.001
		Xanthurenate	↓	-	MS vs. HC	*p* < 1.0 × 10^−4^	FDR < 0.05
Levi (2021) [36]	MS and HC	β-Hydroxyasparagine	↑	-	MS vs. HC	*p* < 0.004	FDR < 0.05
		Sphingosine-1-phosphate (S1P)	↓	-	MS vs. HC	*p* < 0.007	FDR < 0.05
		Carnitine	↓	-	MS vs. HC	*p* < 0.007	FDR < 0.05
		Indolepropionate	↓	-	MS vs. HC	*p* < 0.03	FDR < 0.05
		Indolelactate	↓	-	MS vs. HC	*p* < 0.03	FDR < 0.05
		*p*-Cresol sulfate	↑	-	MS vs. HC	*p* = 0.12	FDR (ns)
		Stachydrine	↑	-	MS vs. HC	NR	FDR < 0.05
		3-Hydroxyhippurate	↑	-	MS vs. HC	NR	FDR < 0.05
Probert (2021) [19]	CIS converters and non-converters	Glucose	↑	-	CIS converters vs. non-converters	*p* < 0.05	FDR (ns)
		Lactate	↑	-	CIS converters vs. non-converters	*p* < 0.05	FDR (ns)
		Myo-inositol	↓	-	CIS converters vs. non-converters	*p* < 0.05	FDR (ns)
		Creatine	↓	-	CIS converters vs. non-converters	*p* < 0.05	FDR (ns)
Keller (2021) [21]	MS and HC (after PRT treatment)	Myristate	↑	-	MS vs. HC after PRT	*p* < 0.001	FDR (ns)
		Palmitate	↑	-	MS vs. HC after PRT	*p* < 0.001	FDR (ns)
		Arachidonate	↑	-	MS vs. HC after PRT	*p* = 0.0012	FDR (ns)
		Oleate/Vaccinate	↑	-	MS vs. HC after PRT	*p* = 0.0012	FDR (ns)
		Linoleate	↑	-	MS vs. HC after PRT	*p* = 0.0017	FDR (ns)
		5-Dodecenoate	↑	-	MS vs. HC after PRT	*p* = 0.003	FDR (ns)
		1-Palmitoleoyl-GPC	↑	-	MS vs. HC after PRT	*p* = 0.00012	FDR (ns)
		1-Stearoyl-GPC	↑	-	MS vs. HC after PRT	*p* = 0.00061	FDR (ns)
		1-1-Enyl-palmitoyl-GPC	↑	-	MS vs. HC after PRT	*p* = 0.0012	FDR (ns)
		Margarate	↑	-	MS vs. HC after PRT	*p* = 0.00061	FDR (ns)
		Dihomolinoleate	↑	-	MS vs. HC after PRT	*p* = 0.0017	FDR (ns)
		Stearidonate	↑	-	MS vs. HC after PRT	*p* = 0.0017	FDR (ns)
		3-Hydroxyisobutyrate	↑	-	MS vs. HC after PRT	*p* = 0.00037	FDR (ns)
		Glycerol	↑	-	MS vs. HC after PRT	*p* = 0.0031	FDR (ns)
		3-Aminoisobutyrate	↑	-	MS vs. HC after PRT	*p* = 0.0052	FDR (ns)
		Hydantoin-5-propionate	↑	-	MS vs. HC after PRT	*p* = 0.00061	FDR (ns)
		DHEAS	↑	-	MS vs. HC after PRT	*p* = 0.03	FDR (ns)
		Acylcarnitines	↑	-	MS vs. HC after PRT	*p* = 0.018	FDR (ns)
		Stearate	↑	-	MS vs. HC after PRT	*p* = 0.0040	FDR (ns)
Yeo (2021) [37]	RRMS	Lysine	↑	-	RRMS, relapse vs. long remission (LR ≥ 24 M)	*p* < 0.05	FDR (ns)
		Asparagine	↑	-	RRMS, relapse vs. long remission (LR ≥ 24 M)	*p* < 0.05	FDR (ns)
		Isoleucine	↓	-	RRMS, relapse vs. long remission (LR ≥ 24 M)	*p* < 0.01	FDR (ns)
		Leucine	↓	-	RRMS, relapse vs. long remission (LR ≥ 24 M)	*p* < 0.01	FDR (ns)
Yeo (2020) [27]	RRMS and SPMS	Lipoproteins	↓	-	SPMS vs. RRMS	NR	FDR < 0.05
		Choline	↓	-	SPMS vs. RRMS	NR	FDR < 0.05
		3-Hydroxybutyrate	↓	-	SPMS vs. RRMS	NR	FDR < 0.05
		Glucose	↑	-	SPMS vs. RRMS	NR	FDR < 0.05
		*N*-acetylated glycoproteins	↑	-	SPMS vs. RRMS	NR	FDR < 0.05
Gaetani (2020) [34]	RRMS and HC	Kynurenine	↓	-	RRMS vs. HC	*p* = 0.01	NR
		K/T ratio	↓	-	RRMS vs. HC	*p* = 0.04	NR
		Tryptophan	↑	-	RRMS vs. HC	*p* = 0.001	NR
		Indole-3-propionic acid	↑	-	RRMS vs. HC	*p* < 0.001	NR
		Indole-3-propionic acid	↑	-	RRMS with recent relapse (<30 days) vs. stable RRMS	*p* = 0.04	NR
		K/A ratio	↓	-	RRMS with recent relapse (<30 days) vs. stable RRMS	*p* = 0.03	NR
		Anthranilate	↑	-	RRMS with recent relapse (<30 days) vs. stable RRMS	*p* = 0.02	NR
Carlsson (2020) [29]	SPMS and HC	Glycine	↑	-	SPMS vs. HC	*p* = 0.016	FDR < 0.05
		ADMA	↑	-	SPMS vs. HC	*p* = 0.009	FDR < 0.05
		PC-O (34:0)	↑	-	SPMS vs. HC	*p* = 0.046	FDR < 0.05
		Hexoses	↑	-	SPMS vs. HC	*p* = 0.010	FDR < 0.05
Duscha (2020) [24]	MS	Propionic acid	↓	-	MS vs. HC	*p* = 0.0016	FDR < 0.05
Penkert (2020) [44]	HC and MS/CIS (Monozygotic twins)	Ether phosphatidylcholines (PC O-)	↓	-	MS vs. HC co-twins	*p* = 0.00081	FDR = 0.0095
		Ether phosphatidylethanolamines (PE O-)	↓	-	MS vs. HC co-twins	*p* = 0.0015	FDR = 0.0095
		Phosphatidylcholines (PC)	↓	-	MS vs. HC co-twins	*p* = 0.017	FDR = 0.074
		PC O- species with DPA (C22:5)	↓	-	MS vs. HC co-twins	*p* = 0.00047	FDR = 0.011
		PC O- species with other PUFA acyl chains (C22:4, C20:3, C20:4)	↓	-	MS vs. HC co-twins	*p* < 0.05	FDR (ns)
		PC O- species with ether-bound alkyl chains (O-16:0;0, O-16:1;0, O-18:1;0)	↓	-	MS vs. HC co-twins	*p* < 0.05	FDR (ns)
		PC O-16:1;0/20:3;0	↓	-	MS vs. HC co-twins	*p* = 0.00006	FDR = 0.015
		Multiple PC O- species with PUFA side chains	↓	-	MS vs. HC co-twins	*p* < 0.05	FDR (ns)
Amatruda (2020) [31]	PPMS, HC, validation cohort: SPMS and RRMS	DiHexCer(d18:1/18:2)	↓	-	PPMS vs. HC	*p* < 0.05	FDR (ns)
		DiHexCer(d18:1/18:3)	↓	-	PPMS vs. HC	*p* < 0.05	FDR (ns)
		SM(d18:1/14:0)	↓	-	PPMS vs. HC	*p* < 0.01	FDR (ns)
		MonoHexCer(d18:1/20:0)	↑	-	PPMS vs. HC	*p* < 0.01	FDR (ns)
		LPA-18:2	↓	-	PPMS progressors vs. PPMS non-progressors and HC	*p* < 0.05	FDR (ns)
		LPA-18:2	↓	-	SPMS progressors vs. SPMS non-progressors	*p* < 0.05	FDR (ns)
Waddington (2020) [45]	RRMS (IFNβtreated), CIS	VLDL-PA	↑	-	ADA+ vs. ADA−	*p* < 0.05	FDR (ns)
		Cholesteryl esters in VLDL	↑	-	ADA+ vs. ADA−	*p* < 0.05	FDR (ns)
		TG/PG ratio	↑	-	ADA+ vs. ADA−	*p* < 0.05	FDR (ns)
		Free cholesterol	↑	-	ADA+ vs. ADA−	*p* < 0.05	FDR (ns)

**Table 3 ijms-26-09207-t003:** Multivariate and machine-learning approaches in metabolomic studies of MS. This table summarizes studies that applied supervised multivariate or machine-learning methods for metabolomic classification in MS. For each study, the MS phenotype compared, algorithm used (e.g., PLS-DA, SVM, random forest, OPLS-DA), validation strategy (e.g., train–test split, cross-validation, permutation testing), and reported performance metrics (e.g., accuracy, sensitivity, specificity, AUROC) are shown. Cases where key parameters (e.g., AUROC values) were not reported are indicated as NR.

Author (Year)	MS Phenotype	Algorithm	Validation Strategy	Performance Metrics
Datta (2024) [26]	RRMS vs. Progressive	PLS-DA, SVM, Random Forest (biosigner feature selection/classification)	Train–test split; classifiers evaluated on independent test set; feature tiers assigned by biosigner.	Prediction accuracy reported; AUROC NR.
Yeo (2020) [27]	RRMS vs. SPMS	OPLS-DA	External 10-fold cross-validation with repetition and permutation testing; ensemble of 1000 models; independent test sets	Accuracy, sensitivity, specificity reported

## Data Availability

Not applicable.

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
