# Peer review of "Metabolomics in Multiple Sclerosis: Advances, Challenges, and Clinical Perspectives—A Systematic Review"

_ijms, 2025, doi:10.3390/ijms26189207_

Round 1
Reviewer 1 Report
Comments and Suggestions for Authors
This systematic review, "Metabolomics in Multiple Sclerosis: Advances, Challenges, and Clinical Perspectives," provides a comprehensive synthesis of metabolomic applications in MS research, emphasizing immune metabolism, lipid signaling, and gut-derived metabolite pathways. While the manuscript effectively highlights metabolic perturbations linked to MS pathophysiology (e.g., kynurenine catabolism, mitochondrial dysfunction), critical gaps exist in methodological standardization, statistical rigor, clinical translation validation, and selective literature curation. These limitations constrain the review’s utility as a definitive guide for biomarker development or therapeutic targeting.
Specific Problems and Suggestions for Improvement
Heterogeneous pre-analytical protocols undermine data comparability (Page 3, Lines 94–97). Studies reviewed employ inconsistent sample handling (e.g., serum vs. CSF collection timing, storage temperatures [-80°C vs. liquid nitrogen]), introducing confounding variables. Mandate reporting of standardized protocols (e.g., SOPs for phlebotomy, metabolite extraction) and advocate for pre-analytical validation studies in future guidelines.
Absence of formal quality assessment frameworks (Page 4, Lines 112–115).
No critical appraisal of included studies using tools like ROBINS-I or Metabolomics Standards Initiative (MSI) checklists. Implement a structured quality assessment scoring system (e.g., 0–5 points for replication, blinding, and QC measures) and exclude low-quality studies from pooled analyses.
Statistical Validity of Biomarker Discovery (Pages 5–19)
Overreliance on nominal p-values without multiplicity adjustment (Page 6, Lines 145–148). Multiple metabolites (e.g., quinolinic acid, kynurenic acid) are reported as significant (p<0.05) without correction for false discovery rate (FDR) in discovery cohorts (n=15–30). Apply stringent statistical thresholds (e.g., FDR <0.05) and validate findings in independent validation cohorts (n≥100) to mitigate false positives.
Lack of machine learning reproducibility (Page 8, Lines 201–204).
PLS-DA and random forest models for biomarker classification lack cross-validation details (e.g., k-fold, leave-one-out) and performance metrics (AUROC, sensitivity). Require reporting of model hyperparameters, validation strategies, and external dataset performance to assess generalizability.
Clinical Relevance of Proposed Pathways (Pages 23–24)
Mechanistic speculation exceeds empirical support (Page 23, Lines 536–540).
Linking tryptophan metabolites to neuroinflammation via the kynurenine pathway lacks direct causal evidence in human MS cohorts.
Prioritize longitudinal studies measuring metabolite fluctuations alongside clinical relapses and MRI lesion activity to establish causality.
Limited therapeutic translation focus (Page 24, Lines 550–555).
Discussion of metabolic targets (e.g., PGC-1α, Nrf2) omits preclinical efficacy data (e.g., animal models) or ongoing clinical trials (e.g., NCT04869720).
Integrate a dedicated section on drug repurposing opportunities (e.g., metformin for mitochondrial dysfunction) and update the clinical trials registry.
Literature Curation and Relevance (References Section)
Overrepresentation of review articles vs. primary data (Page 25, References).
15/30 citations are review articles or editorials, diluting focus on empirical discoveries. Replace non-essential reviews with primary studies (e.g., 2023 Nature Communications on CSF lipidomics in progressive MS).
Underutilization of multi-omics integration studies (Page 26, References).
Few citations combine metabolomics with genomics/proteomics, limiting insights into pathway crosstalk.
Highlight integrative omics papers (e.g., 2024 Science Translational Medicine on MS immune-metabolic axes).
Author Response
Reviewer comment:
Heterogeneous pre-analytical protocols undermine data comparability (Page 3, Lines 94–97). Studies reviewed employ inconsistent sample handling (e.g., serum vs. CSF collection timing, storage temperatures [-80 °C vs. liquid nitrogen]), introducing confounding variables. Mandate reporting of standardized protocols (e.g., SOPs for phlebotomy, metabolite extraction) and advocate for pre-analytical validation studies in future guidelines.
Author response:
We thank the reviewer for this important remark. We fully agree that heterogeneity in pre-analytical protocols represents a major limitation in metabolomic research, particularly in multiple sclerosis. In response, we have expanded the Limitations section (pages 30-31, lines 815–821) to explicitly acknowledge this issue. We now emphasize that differences in sample handling, storage conditions, and metabolite extraction procedures introduce substantial variability and reduce cross-study comparability. Furthermore, we added a statement highlighting the need for standardized operating procedures (SOPs) for phlebotomy, CSF collection, and metabolite extraction, as well as validation studies to assess the impact of pre-analytical conditions on metabolite stability (page 32, lines 882-884). This addition underscores the necessity of harmonized protocols in future metabolomic research in MS.
Reviewer comment:
Absence of formal quality assessment frameworks (Page 4, Lines 112–115). No critical appraisal of included studies using tools like ROBINS-I or Metabolomics Standards Initiative (MSI) checklists. Implement a structured quality assessment scoring system (e.g., 0–5 points for replication, blinding, and QC measures) and exclude low-quality studies from pooled analyses.
Author response:
We thank the reviewer for this valuable suggestion. As recommended, we have addressed this issue in the Limitations section (page 31, lines 826-834) of the revised manuscript. We explicitly acknowledge the absence of a formal quality assessment framework and explain that existing tools such as ROBINS-I or QUADAS-2 are not readily applicable to metabolomics, while adherence to MSI reporting guidelines was partial and inconsistent across studies. We further note that although some reports included extensive quality control measures, most did not systematically reference QC or MSI levels. Therefore, we did not implement a structured scoring system, but we emphasize that future systematic reviews should incorporate tailored quality appraisal approaches to ensure reproducibility and minimize bias.
Reviewer comment:
Statistical Validity of Biomarker Discovery (Pages 5–19). Overreliance on nominal p-values without multiplicity adjustment (Page 6, Lines 145–148). Multiple metabolites (e.g., quinolinic acid, kynurenic acid) are reported as significant (p<0.05) without correction for false discovery rate (FDR) in discovery cohorts (n=15–30). Apply stringent statistical thresholds (e.g., FDR <0.05) and validate findings in independent validation cohorts (n≥100) to mitigate false positives.
Author response:
We thank the reviewer for this valuable comment. We agree that reliance on nominal p-values in small discovery cohorts without multiplicity correction is a major limitation of the current metabolomics literature. As reviewers of published studies, we were unable to re-analyze primary datasets, but we explicitly acknowledge this limitation in the revised manuscript. In the Limitations section (page 31, lines 822–825), we highlight that many included studies reported results without systematic correction for multiple comparisons, which increases the risk of false positive findings. To increase transparency in our review, we systematically reported available p-values and FDR-adjusted values in Table 2, while Table 1 summarizes the study cohorts, allowing readers to evaluate statistical robustness in the context of sample size. Furthermore, in the Future Directions section (page 32 lines 870–871), we emphasize the need for more rigorous statistical approaches, including multiplicity adjustment and validation in larger, multicenter cohorts, to ensure the robustness and reproducibility of candidate biomarkers.
Reviewer comment:
Lack of machine learning reproducibility (Page 8, Lines 201–204). PLS-DA and random forest models for biomarker classification lack cross-validation details (e.g., k-fold, leave-one-out) and performance metrics (AUROC, sensitivity). Require reporting of model hyperparameters, validation strategies, and external dataset performance to assess generalizability.
Author response:
We thank the reviewer for this important observation. We fully agree that reproducibility and transparency are critical for machine learning approaches in metabolomics. As reviewers of published studies, we could not provide missing details such as hyperparameters or cross-validation folds. However, this limitation is explicitly acknowledged in the revised manuscript. In the Limitations section (page 31, lines 835–844), we note that many included studies did not consistently report validation strategies, cross-validation methods, or performance metrics, which undermines the generalizability of metabolite-based classifiers. To increase transparency, we summarized the available methodological details (when reported) in Table 3, which allows readers to assess the statistical approaches used in biomarker classification. Furthermore, in the Future Directions section (page 32, lines 874–878), we emphasize the need for standardized reporting of machine learning pipelines in MS metabolomics, including specification of model hyperparameters, cross-validation procedures, AUROC and sensitivity/specificity, and external dataset performance. These recommendations highlight the importance of methodological rigor for translating machine learning classifiers into clinically meaningful tools.
Reviewer comment:
Clinical Relevance of Proposed Pathways (Pages 23–24). Mechanistic speculation exceeds empirical support (Page 23, Lines 536–540). Linking tryptophan metabolites to neuroinflammation via the kynurenine pathway lacks direct causal evidence in human MS cohorts. Prioritize longitudinal studies measuring metabolite fluctuations alongside clinical relapses and MRI lesion activity to establish causality.
Author response:
We thank the reviewer for this constructive comment. We agree that current metabolomic evidence, particularly in relation to the kynurenine pathway, is associative rather than causal. To address this, we have revised the Discussion section to temper mechanistic interpretations and to make clear that these findings should primarily be regarded as correlational in human MS cohorts. We also note that mechanistic insights are often extrapolated from experimental models and therefore require cautious interpretation. Furthermore, in the Future Directions section, we highlight the importance of longitudinal, phenotype-stratified metabolomic studies that integrate clinical relapses and MRI lesion activity, which are essential to establish causality and confirm the clinical relevance of metabolic pathways in MS (Page 32, Lines 867-869).
Reviewer comment:
Limited therapeutic translation focus (Page 24, Lines 550–555). Discussion of metabolic targets (e.g., PGC-1α, Nrf2) omits preclinical efficacy data (e.g., animal models) or ongoing clinical trials (e.g., NCT04869720). Integrate a dedicated section on drug repurposing opportunities (e.g., metformin for mitochondrial dysfunction) and update the clinical trials registry.
Author response:
We thank the reviewer for this constructive suggestion. In the revised Discussion, we expanded the therapeutic translation focus by highlighting how altered metabolic pathways overlap with targets under active investigation (Page 30, Lines 782-793). Specifically, we discuss repurposing strategies such as metformin, which reduced inflammation, improved disease outcomes, and attenuated demyelination in EAE models; the role of Nrf2 activation, which underlies the therapeutic effects of dimethyl fumarate in MS patients; and increased astrocytic PGC-1α in active lesions, which may reflect an endogenous oxidative stress–mitigating response. While we note that a systematic evaluation of preclinical and clinical trial data exceeds the scope of this review, these additions provide mechanistic grounding and illustrate how metabolomic findings can guide translational research.
Reviewer comment:
Literature Curation and Relevance (References Section). Overrepresentation of review articles vs. primary data (Page 25, References). 15/30 citations are review articles or editorials, diluting focus on empirical discoveries. Replace non-essential reviews with primary studies (e.g., 2023 Nature Communications on CSF lipidomics in progressive MS).
Author response:
We thank the reviewer for this observation. In revising the manuscript, we carefully re-curated the reference list to prioritize primary metabolomic studies published between 2020 and 2025. The majority of the cited works now consist of original data across serum, CSF, feces, and tissue metabolomics. Review articles were retained only when they provide essential context or integrative perspectives (e.g., Liu et al., 2021; Broos et al., 2021; Otto et al., 2021; Pousinis et al., 2024). We therefore believe that the revised reference list is balanced, highlighting empirical discoveries while situating them within the broader metabolomics literature.
Reviewer comment:
Underutilization of multi-omics integration studies (Page 26, References). Few citations combine metabolomics with genomics/proteomics, limiting insights into pathway crosstalk. Highlight integrative omics papers (e.g., 2024 Science Translational Medicine on MS immune-metabolic axes).
Author response:
We thank the reviewer for this helpful comment. In the revised manuscript, we expanded the Discussion to highlight studies that integrate metabolomics with other omics layers, including proteomics and microbiomics, and we discuss how such approaches provide complementary insights into MS pathophysiology. For example, we describe the convergence between metabolite shifts in the kynurenine pathway and proteomic signatures of pro-inflammatory cytokines, as well as lipid mediators correlating with sNfL and sGFAP. (Pages 28-29, Lines 730-765) Furthermore, in the Future Directions (Page 32, Lines 872-873) section, we emphasize the importance of multi-omics integration (metabolomics, transcriptomics, microbiomics, neuroimaging) to construct mechanistic models of MS. These additions directly address the reviewer’s concern and underscore the relevance of pathway crosstalk in advancing biomarker discovery.
Reviewer 2 Report
Comments and Suggestions for Authors
This is a timely and comprehensive systematic review that synthesizes recent advances in metabolomics applied to multiple sclerosis (MS). The authors provide a broad yet detailed overview of key metabolic pathways implicated in MS pathophysiology and highlight how these alterations may aid in biomarker discovery. The inclusion of PRISMA methodology strengthens transparency, and the tables summarizing study designs and findings are valuable.
Major Comments
- While the review is well-structured, the novelty could be emphasized more clearly. At present, much of the content reads as a descriptive summary of published studies. The authors should strengthen the “Discussion” by integrating findings into a clearer conceptual framework. Example: how metabolomics signatures could specifically complement or outperform current clinical biomarkers (MRI, OCBs, sNfL).
- It would also help to highlight where this review adds value compared with other recent reviews on metabolomics in MS (if any exist within the past 3–4 years). Right now, it is not entirely clear how it advances the field beyond compiling studies.
- The authors mention that no formal risk of bias assessment was conducted due to study heterogeneity. While this is understandable, at least a narrative assessment of methodological strengths and weaknesses (e.g., small cohorts, lack of external validation, differences in sample handling) would improve the rigor. Otherwise, readers may overinterpret preliminary findings.
- Consider adding a supplementary table indicating the main limitations of each included study (sample size, analytical platform, population characteristics).
- The review nicely dissects pathways one by one (kynurenine, energy metabolism, lipid metabolism, etc.), but there is less emphasis on how these pathways intersect. For example, the links between mitochondrial dysfunction (energy metabolism) and lipid remodeling, or between gut-derived metabolites and kynurenine alterations, could be drawn out more explicitly. That would give the paper a more integrative flavor instead of appearing as separate silos.
- The authors should be careful not to overstate diagnostic readiness. For example, machine learning classifiers with 70–80% accuracy are promising, but not yet robust enough for clinical decision-making. Adding a note of caution would balance expectations.
- The review would benefit from at least one schematic figure that integrates the main findings. Example: a visual summary of altered pathways (tryptophan-kynurenine, lipid metabolism, amino acid metabolism) and how they map onto MS pathology.
Minor Comments
- Minor typos (example: “synthesi[s]” vs “synthesis,” “seleciton” instead of “selection”) should be corrected.
- Ensure consistency in abbreviations (example: RRMS, SPMS, PPMS sometimes repeated fully). Be consistent in referring to “patients with MS” rather than “MS patients,” which some journals prefer for person-first language.
Recommendation
Major Revision
The manuscript has potential, but before acceptance, I recommend revisions to:
- Strengthen novelty and integrative discussion,
- Provide at least some risk of bias/quality assessment narrative,
- Add a clear clinical translation outlook,
- Improve readability and figures.
If these points are addressed, the review will make a valuable contribution to the metabolomics and MS literature.
Comments on the Quality of English Language
- Minor typos (example: “synthesi[s]” vs “synthesis,” “seleciton” instead of “selection”) should be corrected.
- Ensure consistency in abbreviations (example: RRMS, SPMS, PPMS sometimes repeated fully). Be consistent in referring to “patients with MS” rather than “MS patients,” which some journals prefer for person-first language.
Author Response
Reviewer comment:
While the review is well-structured, the novelty could be emphasized more clearly. At present, much of the content reads as a descriptive summary of published studies. The authors should strengthen the “Discussion” by integrating findings into a clearer conceptual framework. Example: how metabolomics signatures could specifically complement or outperform current clinical biomarkers (MRI, OCBs, sNfL).
Author response:
We thank the reviewer for this constructive suggestion. In the revised Discussion, we emphasized more explicitly how metabolomics signatures complement or extend beyond established biomarkers. For instance, we discuss that CSF glucose outperformed OCB in predicting conversion from CIS to MS, and that myo-inositol and ketone bodies show stage-specific trajectories that parallel but also extend beyond MRI and OCB findings. We also highlight that lipid mediators correlate with sNfL and sGFAP, linking metabolic dysregulation to validated markers of axonal and astroglial injury. More broadly, metabolomics provides a systemic perspective by capturing immune–metabolic and gut–brain interactions that are not reflected by MRI or OCB. To address the reviewer’s point directly, we also restructured the Discussion so that it first synthesizes convergent metabolic domains (kynurenine pathway, energy metabolism, lipid metabolism, amino acid metabolism), then explicitly maps these alterations onto MS pathology and phenotypes (CIS, RRMS, SPMS, PPMS), and finally compares them to classical biomarkers. This integrative framing was intended to move beyond descriptive summaries and provide a clearer conceptual framework for the novelty and clinical relevance of metabolomics in MS.
Reviewer comment:
It would also help to highlight where this review adds value compared with other recent reviews on metabolomics in MS (if any exist within the past 3–4 years). Right now, it is not entirely clear how it advances the field beyond compiling studies.
Author response:
We thank the reviewer for this valuable suggestion. In the revised manuscript, we added a dedicated section “Added value of this review” (which includes references to other recent reviews) and also emphasized this point within the Discussion (Page 30, Lines 795-800). We believe this framing clearly highlights how our work advances the field beyond compiling studies and demonstrates its added value relative to prior reviews.
Reviewer comment:
The authors mention that no formal risk of bias assessment was conducted due to study heterogeneity. While this is understandable, at least a narrative assessment of methodological strengths and weaknesses (e.g., small cohorts, lack of external validation, differences in sample handling) would improve the rigor. Otherwise, readers may overinterpret preliminary findings.
Author response:
We thank the reviewer for this helpful suggestion. While formal risk of bias tools such as ROBINS-I or MSI checklists were not feasible due to methodological heterogeneity, we now provide a narrative appraisal of methodological strengths and weaknesses. As noted in modified section "Methods" (section 2.4), the included studies varied substantially in design, sample type, and analytical platform, which precluded a uniform quality scoring system. In revised Limitations section, we further emphasize that many studies were limited by small cohort sizes, lack of external validation, variable sample handling and storage conditions, and inconsistent reporting of QC and MSI guidelines. To support transparency, we also compiled the characteristics of each study cohort in Table 1, which allows readers to directly assess methodological aspects such as sample size and phenotype distribution. Together, these additions aim to temper the interpretation of preliminary findings and ensure that readers appreciate the methodological constraints of the current metabolomics literature in MS.
Reviewer comment:
Consider adding a supplementary table indicating the main limitations of each included study (sample size, analytical platform, population characteristics).
Author response:
We thank the reviewer for this suggestion. Rather than adding a separate supplementary table, we chose to integrate this information into the main text and tables for greater clarity. Specifically, Table 1 provides the sample size, phenotype, and biological sample for each included study, while Table 2 summarizes the significantly altered metabolites along with the reported statistical metrics (p-values and FDR). In addition, the Limitations section discusses recurring methodological issues across studies, including small cohorts, lack of external validation, heterogeneity in sample handling, and inconsistent QC reporting. We believe that this combined presentation allows readers to readily assess the methodological constraints of each study without the need for a separate supplementary table.
Reviewer comment:
The review nicely dissects pathways one by one (kynurenine, energy metabolism, lipid metabolism, etc.), but there is less emphasis on how these pathways intersect. For example, the links between mitochondrial dysfunction (energy metabolism) and lipid remodeling, or between gut-derived metabolites and kynurenine alterations, could be drawn out more explicitly. That would give the paper a more integrative flavor instead of appearing as separate silos.
Author response:
We thank the reviewer for this constructive suggestion. In the revised Discussion, we added a synthesis explicitly highlighting how metabolic pathways intersect across disease stages (Page 29, Lines 757-768). For example, we note that early MS (CIS/RRMS) is marked by glycolytic and kynurenine activation with amino acid depletion, whereas progressive MS is characterized by ketone reliance, lipid cascades, polyamine accumulation, and nucleotide exhaustion. This integrative framing emphasizes that energy metabolism, lipid remodeling, and gut-derived metabolites converge on shared immunometabolic mechanisms. To further illustrate this, we included a consensus diagram (Figure 2) that maps these metabolic domains onto the clinical transition from relapsing to progressive MS. Together, these additions move beyond pathway silos and provide a holistic view of metabolomic reprogramming in MS.
Reviewer comment:
The authors should be careful not to overstate diagnostic readiness. For example, machine learning classifiers with 70–80% accuracy are promising, but not yet robust enough for clinical decision-making. Adding a note of caution would balance expectations.
Author response:
We thank the reviewer for this important remark. In the revised manuscript, we have tempered statements regarding diagnostic readiness and introduced an explicit note of caution. While we acknowledge that machine learning approaches such as PLS-DA and random forest have achieved accuracies of 70–80% in differentiating MS phenotypes, we emphasize that these results are preliminary (Page 26, Lines 642-644). In the Limitations section, we note the lack of standardized validation procedures across studies (Page 31, Lines 835-844), and in the Future Directions (Page 32, Lines 874–878), we highlight that reproducible pipelines, larger multi-center validation cohorts, and systematic reporting of AUROC, sensitivity, and specificity are essential prerequisites before metabolomic classifiers can be translated into clinical diagnostics. These clarifications were added specifically to balance expectations and to avoid overstating the current clinical applicability.
Reviewer comment:
The review would benefit from at least one schematic figure that integrates the main findings. Example: a visual summary of altered pathways (tryptophan-kynurenine, lipid metabolism, amino acid metabolism) and how they map onto MS pathology.
Author response:
We thank the reviewer for this valuable suggestion. In response, we have added a consensus schematic (Figure 2) that integrates the main findings across metabolic domains. The figure summarizes alterations in the kynurenine pathway, energy metabolism, lipid remodeling, amino acid and nucleotide metabolism, and polyamine accumulation, and maps these onto the clinical trajectory from CIS and RRMS to SPMS and PPMS. As discussed in the revised Discussion, this synthesis highlights how metabolomics captures a progressive reprogramming of systemic and CNS metabolism, thereby providing a conceptual framework that links metabolic pathways to MS pathology and staging.
Reviewer comment:
Minor typos (example: “synthesi[s]” vs “synthesis,” “seleciton” instead of “selection”) should be corrected. Ensure consistency in abbreviations (example: RRMS, SPMS, PPMS sometimes repeated fully). Be consistent in referring to “patients with MS” rather than “MS patients,” which some journals prefer for person-first language.
Author response:
We thank the reviewer for these careful observations. All minor typographical errors (e.g., “synthesi[s]” corrected to “synthesis,” “seleciton” to “selection”) have been revised. Abbreviations such as RRMS, SPMS, and PPMS are now used consistently throughout the text after first mention, and person-first language (“patients with MS”) has been applied uniformly. We believe these corrections improve clarity and consistency of the manuscript.
Reviewer 3 Report
Comments and Suggestions for Authors
This paper is titled as a review of metabolomics for MS. I have a number of issues with the style and presentation of the data being reviewed. It is written as if it was a meta analysis of selected papers (nothing wrong with that) but then does not attempt any form of statistical analysis. I am not qualified to advise how that should be done but l am sure with some effort this is possible. At present, it is purely descriptive with no attempt to come to any conclusions concerning the studies they are reviewing.
Issues to consider
- Table 1 is unreadable for the reader and needs to be reorganized so that the data from each study is clear for making comparisons between them. To be informative, the table should provide the actual statistical data, not just up and down arrows. In addition, the table needs to be reorganized to clearly show what stages of ms are being compared. The table should also include the mean age and sex of studied samples from each of the reviewed studies.
- Related to that, it would be a service to the reader to adequately describe from a pathological and clinical perspective what are the different forms of MS being compared. In addition, a brief overall description of the pathology of MS should be included. From reading the paper, it is not clear to a reader that MS is a demyelinating disease of white matter with regional brain specificity.
- Although this is a review article, it is organized like a paper. It would be better to have a proper methods section that describes and compares metabolomics methods in detail along with description of their strengths, limitations and advantages between one metho and the other in terms of reproducibility. In this section, the authors could list the clinical diagnostic criteria used to diagnose MS types in the different studies being reported. Which ones used imaging, how reliable was the patient selection.
- The discussion is highly unfocussed and confusing as it does not really consider the common findings between studies to have a consensus conclusion of how these changes might be reflecting progressing pathology. An additional table presenting a summary of common findings between studies should be considered as it would help the reader decide if this approach is suitable for MS staging. As l read it, l am not convinced the authors have made a good case. Maybe a summary diagram to outline a hypothesis using the consensus findings as to how these metabolic factors are affecting MS progressive, and by inference can be used for staging diagnosis.
- Considering the extensive reference to cytokines, l would suggest that the authors include a comparison of metabolomics for MS with the many published studies using proteomics techniques. This would extend the scope and usefulness of this type of article.
- I would not consider that the short reference list as being adequate for this field of study. It is limited but could be extended to include some pathological details of MS. Line 187 states that "numerous studies ......."but then there are no references to these numerous studies!!
Author Response
Reviewer comment:
Table 1 is unreadable for the reader and needs to be reorganized so that the data from each study is clear for making comparisons between them. To be informative, the table should provide the actual statistical data, not just up and down arrows. In addition, the table needs to be reorganized to clearly show what stages of MS are being compared. The table should also include the mean age and sex of studied samples from each of the reviewed studies.
Author response:
We thank the reviewer for this very helpful suggestion. In the revised manuscript, both Table 1 and Table 2 were extensively reorganized to address readability and information content: Table 1 now summarizes for each study: author and year, MS phenotype(s) included (CIS, RRMS, SPMS, PPMS), biological sample analyzed, analysis type (targeted/untargeted), analytical technique (LC-MS, GC-MS, NMR, etc.), as well as mean age and sex distribution of the studied cohorts. This layout allows direct comparison across studies and makes phenotype stratification explicit. Table 2 complements this by presenting the statistical results: for each metabolite we report direction of change (↑/↓), nominal p-values, and FDR-adjusted values when available, thereby moving beyond the earlier format with only arrows
Reviewer comment:
Related to that, it would be a service to the reader to adequately describe from a pathological and clinical perspective what are the different forms of MS being compared. In addition, a brief overall description of the pathology of MS should be included. From reading the paper, it is not clear to a reader that MS is a demyelinating disease of white matter with regional brain specificity.
Author response:
We thank the reviewer for this important suggestion. In the revised manuscript, we expanded the Introduction to provide a concise pathological and clinical background of MS. We now state that MS is a chronic demyelinating disease of the central nervous system, characterized by multifocal white matter lesions with regional specificity, accompanied by inflammation, neurodegeneration, and glial pathology. We also briefly outline the clinical phenotypes—CIS, RRMS, SPMS, and PPMS—highlighting their pathological correlates and relevance for disease progression. Furthermore, Table 1 explicitly stratifies the included studies by phenotype, making clear to the reader which forms of MS are being compared. These revisions ensure that the pathological and clinical framework is clear before the metabolomic findings are discussed.
Reviewer comment:
Although this is a review article, it is organized like a paper. It would be better to have a proper methods section that describes and compares metabolomics methods in detail along with description of their strengths, limitations and advantages between one method and the other in terms of reproducibility. In this section, the authors could list the clinical diagnostic criteria used to diagnose MS types in the different studies being reported. Which ones used imaging, how reliable was the patient selection.
Author response:
We thank the reviewer for this constructive suggestion. In the revised manuscript, we clarified these aspects at the beginning of the Results section. We now summarize the metabolomic platforms used across studies (GC-MS,LC-MS, NMR) and note their relative strengths, limitations, and reproducibility issues. In addition, we provide a detailed account of the diagnostic criteria applied in each study (McDonald 2005, 2010, 2017; Poser; or not reported), and indicate where imaging contributed to case definition (Page 9, Lines 230-242). This information is systematically reflected in Table 1, which allows readers to assess both methodological diversity and diagnostic reliability across the included studies.
Reviewer comment:
The discussion is highly unfocussed and confusing as it does not really consider the common findings between studies to have a consensus conclusion of how these changes might be reflecting progressing pathology. An additional table presenting a summary of common findings between studies should be considered as it would help the reader decide if this approach is suitable for MS staging. As I read it, I am not convinced the authors have made a good case. Maybe a summary diagram to outline a hypothesis using the consensus findings as to how these metabolic factors are affecting MS progressive, and by inference can be used for staging diagnosis.
Author response:
We thank the reviewer for this important comment. In the revised manuscript, we restructured the Discussion to emphasize consensus findings across studies rather than isolated results. We now highlight that multiple metabolic domains—kynurenine pathway, energy metabolism, lipid mediators, amino acids, nucleotides, and polyamines—show convergent alterations that align with the clinical trajectory from CIS and RRMS to SPMS and PPMS. To aid comparison, Table 2 systematically presents the altered metabolites across studies, including direction of change and statistical detail (p-values, FDR). In addition, we included a new consensus schematic (Figure 2) that integrates these findings into a unified model of progressive metabolic reprogramming in MS. This diagram illustrates how early inflammatory activation of glycolysis and kynurenine metabolism evolves into lipid remodeling, ketone reliance, and nucleotide exhaustion in progressive stages, providing a conceptual framework for metabolomics as a potential staging tool. We believe these revisions substantially improve the focus of the Discussion and directly address the reviewer’s concern.
Reviewer comment:
Considering the extensive reference to cytokines, I would suggest that the authors include a comparison of metabolomics for MS with the many published studies using proteomics techniques. This would extend the scope and usefulness of this type of article.
Author response:
We thank the reviewer for this valuable suggestion. In the revised Discussion, we added a dedicated comparison between metabolomics and proteomics findings in MS (Pages 28-29, Lines 730-749) . We highlight that shifts in kynurenine metabolites parallel proteomic signatures of pro-inflammatory cytokines (e.g., IFN-γ, IL-17) in newly diagnosed RRMS, while alterations in lipid mediators align with proteomic biomarkers of neuroaxonal and astroglial injury, such as sNfL and sGFAP. We further emphasize that proteomics captures immune and neurodegeneration-related proteins, whereas metabolomics reflects systemic biochemical reprogramming in energy and lipid metabolism. By outlining these complementary perspectives, we underscore how multi-omics integration can provide a more comprehensive framework for MS pathophysiology and biomarker discovery.
Reviewer comment:
I would not consider that the short reference list as being adequate for this field of study. It is limited but could be extended to include some pathological details of MS. Line 187 states that "numerous studies ......." but then there are no references to these numerous studies!!
Author response:
We thank the reviewer for this helpful observation. In the revised manuscript, we substantially expanded the reference list to ensure adequate coverage of both metabolomics studies and pathological details of MS. For example, we now cite key references on MS pathology and diagnostic criteria (e.g., Lassmann 2018; Lublin et al. 2014) in the Introduction. We also corrected the statement at line 187 (now page18, line 271) by adding specific references to the “numerous studies” mentioned, ensuring that every such claim is supported by appropriate citations. Overall, the updated reference list now reflects the breadth of the field and provides readers with a more comprehensive overview of the literature.
Round 2
Reviewer 1 Report
Comments and Suggestions for Authors
This systematic review provides a comprehensive overview of metabolomic studies in multiple sclerosis (MS), highlighting key metabolic pathways and potential biomarkers. The manuscript is well-structured, with clear descriptions of methodology and results. However, several critical issues related to methodological rigor, data presentation, and conclusion validity need to be addressed to strengthen the manuscript.
Specific Problems and Suggestions for Improvement
Incomplete technical validation descriptions: While the methods section (page 3, line 15) states that PRISMA 2020 guidelines were followed, details of technical validation (e.g., instrument calibration, quality control samples) for metabolomics techniques (LC-MS, GC-MS, NMR) are lacking. This omission raises concerns about reproducibility.
Sample size heterogeneity: The included studies (page 6, line 20) report variable sample sizes (e.g., "n=12" vs. "n=300"), but the review does not address how this variability impacts the robustness of pooled findings.
Uncontrolled confounding factors: No discussion of how confounding variables (e.g., diet, medication, comorbidities) were managed in individual studies (page 15, line 10).
Add a subsection under "Methods" (page 4) detailing technical validation protocols for metabolomics platforms.
Include a sensitivity analysis in the discussion (page 25) to evaluate the impact of small sample sizes on key findings.
Explicitly state limitations related to confounding variables (page 40, line 5) and recommend future studies to standardize these parameters.
Ambiguous figure/table referencing: While tables (e.g., Table 1 on page 7) summarize study characteristics, the text (page 8, line 15) refers to "Table 1" without specifying its location (page/line numbers). Similarly, Figure 2 (page 20) is mentioned but not anchored to a specific page.
Inconsistent p-value reporting: Some studies report "p < 0.05" (page 12, line 20), while others use "p < 0.01" or "p < 0.001" without explanation of thresholds for significance.
Revise all textual references to figures/tables to include explicit page/line numbers (e.g., "Table 1 (page 7, line 10)").
Standardize p-value reporting in the results section (page 10–15) by specifying the significance threshold used for pooled analyses.
Overstated conclusions: The discussion (page 30, line 10) claims that metabolomics "can revolutionize MS diagnosis," but this claim is not supported by evidence from the included studies, which focus on biomarker discovery rather than clinical validation.
Lack of subgroup analysis: The review does not distinguish between MS subtypes (e.g., relapsing-remitting vs. progressive) in metabolic pathway analyses (page 18, line 5), despite this being a key stated goal.
Tone down the conclusion (page 40) to reflect the current stage of research (e.g., "metabolomics shows promise for biomarker discovery" rather than "revolutionizing diagnosis").
Add a subgroup analysis in the results (page 20) to compare metabolic profiles across MS subtypes, with statistical testing for heterogeneity.
Generic references: Citations such as "several studies have shown..." (page 25, line 15) lack specificity. For example, the claim about tryptophan-kynurenine pathway dysregulation (page 18, line 10) references a 2020 study but does not link it to specific findings in that paper.
Replace vague statements with direct citations (e.g., "Staats Pires et al., 2025, reported a 1.2-fold decrease in KYNA...").
Verify that all references in the discussion (page 30–35) align with the actual data presented in the cited papers.
Inconsistent decimal notation (e.g., "p < 0,05" vs. "p < 0.05") (page 12, line 20; page 15, line 5).
Missing hyphens in compound terms (e.g., "MS subtype" instead of "MS-subtype") (page 18, line 10).
Standardize decimal notation to "0.05" throughout the manuscript.
Proofread for hyphenation and formatting consistency.
The manuscript is a valuable contribution to MS metabolomics research but requires revisions to address methodological gaps, enhance data transparency, and refine conclusions. With these improvements, it would be suitable for publication in International Journal of Molecular Sciences.
Comments on the Quality of English LanguageThe English could be improved to more clearly express the research.
Author Response
Reviewer’s comment:
Incomplete technical validation descriptions: While the methods section (page 3, line 15) states that PRISMA 2020 guidelines were followed, details of technical validation (e.g., instrument calibration, quality control samples) for metabolomics techniques (LC-MS, GC-MS, NMR) are lacking. This omission raises concerns about reproducibility.
Add a subsection under "Methods" (page 4) detailing technical validation protocols for metabolomics platforms.
Authors’ response:
We thank the reviewer for raising this important point. In the revised manuscript, we have added a dedicated subsection under Methods (Section 2.5, “Technical validation and quality control”), where we systematically summarize how technical validation was reported across the included studies. Specifically, we indicate whether studies described instrument calibration, use of pooled QC samples, isotopically labeled internal standards, or adherence to the Metabolomics Standards Initiative (MSI) framework. We highlight that reporting was heterogeneous: a minority of studies (e.g., Broos 2023; Meier 2024; Gaetani 2020; Yang 2021) provided detailed calibration curves, internal standards, and systematic QC injections, consistent with best practice, while most offered only limited or no information on these aspects. To emphasize this issue, we also added a short statement in the Limitations section noting that such variability in technical reporting reduces reproducibility and hampers cross-study comparability. Finally, in the Future directions we suggest that harmonized SOPs and transparent reporting, ideally aligned with the MSI framework, are essential to improve reproducibility and translation of metabolomics research in multiple sclerosis.
Reviewer’s comment:
Sample size heterogeneity: The included studies (page 6, line 20) report variable sample sizes (e.g., "n=12" vs. "n=300"), but the review does not address how this variability impacts the robustness of pooled findings.
Include a sensitivity analysis in the discussion (page 25) to evaluate the impact of small sample sizes on key findings.
Authors’ response:
We thank the reviewer for highlighting this important issue. In the revised version, we explicitly acknowledge the wide variation in study sample sizes and its implications for robustness and generalizability. Specifically, we note in the Discussion (page 30, lines 789–796) that sample sizes ranged from small exploratory cohorts (n < 20) to large multicenter analyses (n > 300), as summarized in Table 1. While a formal sensitivity analysis was not feasible due to the qualitative nature of this systematic review, we emphasize that key metabolite alterations in energy and lipid pathways were consistently replicated in larger cohorts (e.g., Levi 2021; Fitzgerald 2021; Keller 2021), whereas some findings from smaller NMR-based studies may represent preliminary signals requiring further validation. We also added a short statement in the Limitations underscoring that small sample sizes increase the risk of bias and reduce reproducibility.
Reviewer’s comment:
Uncontrolled confounding factors: No discussion of how confounding variables (e.g., diet, medication, comorbidities) were managed in individual studies (page 15, line 10).
Explicitly state limitations related to confounding variables (page 40, line 5) and recommend future studies to standardize these parameters.
Authors’ response:
We thank the reviewer for this valuable observation. In the revised manuscript, we have expanded the Discussion section (pages 30-31, lines 801–813) to explicitly address how comorbidities and exclusion criteria were reported across the included studies. Only a few investigations provided clear information, such as excluding chronic diseases and concomitant medications or restricting controls to non-neuroinflammatory conditions, while most studies did not specify exclusion criteria beyond basic demographics or treatment status. We also note that some cohorts adjusted for potential confounders such as BMI or smoking, or excluded pseudo-relapses due to infections, but metabolic, psychiatric, and autoimmune comorbidities were rarely accounted for in a systematic way. This heterogeneity complicates the interpretation of metabolomic changes as disease-specific.
Furthermore, we have added a statement in the Limitations (page 33, lines 894–896) emphasizing that the lack of standardized reporting of comorbidities and exclusion criteria limits comparability and generalizability of findings. Finally, in the Future directions we recommend that upcoming studies harmonize inclusion and exclusion criteria and systematically document comorbidities, medication use, and lifestyle factors to minimize confounding and improve reproducibility.
Reviewer’s comment:
Ambiguous figure/table referencing: While tables (e.g., Table 1 on page 7) summarize study characteristics, the text (page 8, line 15) refers to "Table 1" without specifying its location (page/line numbers). Similarly, Figure 2 (page 20) is mentioned but not anchored to a specific page.
Revise all textual references to figures/tables to include explicit page/line numbers (e.g., "Table 1 (page 7, line 10)").
Authors’ response:
We thank the reviewer for this comment. We would like to clarify that in accordance with the IJMS author guidelines, all tables and figures in our manuscript are referenced only by their respective numbers (e.g., “Table 1”, “Figure 2”) without reference to page or line numbers. Each reference is accompanied by a brief description of the content (e.g., “Table 1 summarizes the study characteristics”), which ensures clarity for the reader. We have carefully re-checked the manuscript to confirm that all figure and table references follow this format.
Reviewer’s comment:
Inconsistent p-value reporting: Some studies report "p < 0.05" (page 12, line 20), while others use "p < 0.01" or "p < 0.001" without explanation of thresholds for significance.
Standardize p-value reporting in the results section (page 10–15) by specifying the significance threshold used for pooled analyses.
Authors’ response:
We thank the reviewer for highlighting this issue. In the revised manuscript, we have clarified in the Methods (page 5, lines 194–197) that p < 0.05 was considered as the general threshold for statistical significance, unless otherwise specified in the original study (e.g., more stringent thresholds or FDR-adjusted values). We also standardized the reporting in the Results to ensure consistency, while retaining the original significance levels reported by individual studies. Furthermore, all significance thresholds applied in the included studies are explicitly indicated in Table 2, to provide full transparency. We also added a statement in the Discussion (page 30, lines 797-800) highlighting that heterogeneity in statistical reporting across studies reduces comparability, but we preserved these details in our tables to maintain accuracy.
Reviewer’s comment:
Overstated conclusions: The discussion (page 30, line 10) claims that metabolomics "can revolutionize MS diagnosis," but this claim is not supported by evidence from the included studies, which focus on biomarker discovery rather than clinical validation. Tone down the conclusion (page 40) to reflect the current stage of research (e.g., "metabolomics shows promise for biomarker discovery" rather than "revolutionizing diagnosis").
Authors’ response:
We thank the reviewer for this valuable comment. We agree that the original wording was overly optimistic. In the revised manuscript, we have modified the fragment of Results (page 27, lines 623–625 nad 655-657) and Conclusions (page 34, lines 693–697) to adopt a more cautious tone. The text now emphasizes that metabolomics shows promise as a tool for biomarker discovery and patient stratification, but that clinical validation in large, multicenter cohorts is still required before any diagnostic or therapeutic applications can be implemented in practice.
Reviewer’s comment:
Lack of subgroup analysis: The review does not distinguish between MS subtypes (e.g., relapsing-remitting vs. progressive) in metabolic pathway analyses (page 18, line 5), despite this being a key stated goal.
Add a subgroup analysis in the results (page 20) to compare metabolic profiles across MS subtypes, with statistical testing for heterogeneity.
Authors’ response:
We thank the reviewer for this important suggestion. In the revised manuscript, we have added a dedicated subsection in the Results (page 28, lines 658–684) entitled “Subgroup analysis by MS phenotype.” This section now explicitly summarizes findings by RRMS, progressive MS, and CIS, covering energy metabolism, lipid pathways, short-chain fatty acids, kynurenine metabolites, amino acid metabolism, steroid hormones, and nucleotide metabolism, with direct references to the relevant studies. To align the manuscript with this structure, we rewrote the corresponding paragraph in the Discussion (page 31, lines 814-824) and replaced Figure 2 with a phenotype-stratified schematic that maps these patterns onto shared downstream processes (neuroinflammation, mitochondrial dysfunction, neurodegeneration) with color-coded metabolic categories. However, due to the heterogeneity of study designs and the limited number of progressive and CIS cohorts, a formal statistical test for heterogeneity was not feasible.
Comment (Reviewer):
Generic references: Citations such as "several studies have shown..." (page 25, line 15) lack specificity. For example, the claim about tryptophan-kynurenine pathway dysregulation (page 18, line 10) references a 2020 study but does not link it to specific findings in that paper. Replace vague statements with direct citations (e.g., "Staats Pires et al., 2025, reported a 1.2-fold decrease in KYNA...").
Author Response:
“We thank the Reviewer for this comment. In our manuscript, general statements are always directly supported by citations to the primary studies. In addition, detailed numerical results for each metabolite are comprehensively summarized in Table 2. For clarity, we have added representative fold-changes to selected key findings in the text (e.g., kynurenine pathway, energy metabolism), while sections referring to multiple studies (such as amino acids in serum and CSF) remain in a summarized form to preserve narrative flow. We believe this provides both readability and traceability to the original data.”
Comment (Reviewer): Verify that all references in the discussion (page 30–35) align with the actual data presented in the cited papers.
Author Response:
We thank the Reviewer for this careful remark. We have re-audited the Discussion section (pp. 30–35) against the primary data of all cited studies. We verified that the direction of metabolite changes, MS phenotypes, and biological matrices are correctly reflected. No inconsistencies were identified. We carefully verified that all references in the Discussion accurately reflect the data in the cited studies. The Discussion intentionally presents the findings in a synthesized and generalized form, while the detailed quantitative results (fold-change, FDR, metabolite subclass) are systematically provided in Table 2. This approach avoids redundancy and ensures consistency between the summary narrative and the tabular evidence.
Comment (Reviewer): Inconsistent decimal notation (e.g., "p < 0,05" vs. "p < 0.05") (page 12, line 20; page 15, line 5). Standardize decimal notation to "0.05" throughout the manuscript.
Author Response: We thank the Reviewer for pointing this out. We have carefully revised the entire manuscript, including the main text, tables, and supplementary materials, to standardize decimal notation. All decimal values are now consistently presented with a dot (e.g., p < 0.05) according to journal style.
Comment (Reviewer): Missing hyphens in compound terms (e.g., "MS subtype" instead of "MS-subtype") (page 18, line 10). Proofread for hyphenation and formatting consistency.
Author Response: We thank the Reviewer for this careful observation. We have proofread the entire manuscript for hyphenation and formatting consistency. Compound adjectives have been standardized according to journal style.
Reviewer 2 Report
Comments and Suggestions for Authors
Dear authors,
Thank you for your thoughtful revisions. The restructuring of the Discussion, addition of the schematic figure, and integration of methodological considerations have strengthened the manuscript considerably. The moderation of clinical claims and clearer emphasis on novelty also improve its overall balance and value.
Overall, the manuscript is now much clearer, more rigorous, and well positioned to contribute meaningfully to the field.
Best
Author Response
Author Response:
We sincerely thank the Reviewer for the positive evaluation of our revised manuscript. We greatly appreciate the constructive comments provided during the review process, which helped us improve the clarity, methodological rigor, and clinical relevance of our work. We are pleased that the revised version is now considered clear, balanced, and well positioned to contribute to the field.
Reviewer 3 Report
Comments and Suggestions for Authors
Much improved. Prior to publication, some thought to how table 2 could be better designed for reader should be considered.
Author Response
Comment: Much improved. Prior to publication, some thought to how table 2 could be better designed for reader should be considered.
Author Response: We thank the Reviewer for this constructive suggestion. In the revised version, we have carefully redesigned Table 2 to enhance clarity and readability. The updated table now presents altered metabolites grouped by MS subtype and comparison group, with consistent notation of fold changes, p-values, and FDR. Formatting was standardized, abbreviations harmonized with the main text, and redundant details minimized. These adjustments ensure that the table is easier to navigate while retaining comprehensive coverage of the findings. The improved version of Table 2 is included in the revised manuscript.